# Precise Parameter Localization for Textual Generation in Diffusion Models

**Łukasz Staniszewski*& Bartosz Cywiński***
Warsaw University of Technology
{luks.staniszewski,bcywinski11}@gmail.com

**Franziska Boenisch**
CISPA Helmholtz Center for Information Security
boenisch@cispa.de

**Kamil Deja**
Warsaw University of Technology
IDEAS NCBR
kamil.deja@pw.edu.pl

**Adam Dziedzic**
CISPA Helmholtz Center for Information Security
adam.dziedzic@cispa.de

## Abstract

Novel diffusion models can synthesize photo-realistic images with integrated high-quality text. Surprisingly, we demonstrate through attention activation patching that only less than 1% of diffusion models' parameters, all contained in attention layers, influence the generation of textual content within the images. Building on this observation, we improve textual generation efficiency and performance by targeting cross and joint attention layers of diffusion models. We introduce several applications that benefit from localizing the layers responsible for textual content generation. We first show that a LoRA-based fine-tuning solely of the localized layers enhances, even more, the general text-generation capabilities of large diffusion models while preserving the quality and diversity of the diffusion models' generations. Then, we demonstrate how we can use the localized layers to edit textual content in generated images. Finally, we extend this idea to the practical use case of preventing the generation of toxic text in a cost-free manner. In contrast to prior work, our localization approach is broadly applicable across various diffusion model architectures, including U-Net (e.g., SDXL and DeepFloyd IF) and transformer-based (e.g., Stable Diffusion 3), utilizing diverse text encoders (e.g., from CLIP to the large language models like T5). Project page available at https://t2i-text-loc.github.io/.

## 1 Introduction

Recent advancements in generative models for the vision domain have demonstrated remarkable efficacy in image synthesis tasks and significant improvements in the quality and diversity of the generated outputs (DDPM (Ho et al., 2020), LDM (Rombach et al., 2022)). The next generation of models, including DeepFloyd IF (StabilityAI, 2023), Imagen (Saharia et al., 2022), Stable Diffusion 3 (SD3) (Esser et al., 2024), and FLUX (Labs, 2024), extend this progress to photo-realistic generations with *high-quality visual text*. While introducing impressive capabilities, such models usually operate as black-boxes with complex architectures entangling various skills.

In this work, we propose to shed some light on the inner workings of recent diffusion models and introduce the first method to localize parts of the model responsible for the generation of textual content, based on activation patching technique (Meng et al., 2022). We determine that only $0.61\%$ of Stable Diffusion XL (Podell et al., 2024), $0.21\%$ of Deepfloyd IF (StabilityAI, 2023), and $0.23\%$ of Stable Diffusion 3 (Esser et al., 2024) parameters are responsible solely for this task. Our observations hold across various DMs' architectures, both U-Net and Transformer-based, for DMs utilizing diverse text encoders, such as CLIP (Radford et al., 2021) and T5 (Raffel et al., 2020; Roberts et al., 2022). Additionally, we present several applications that benefit from our localization method.

---

*\***Equal Contribution.** Work done while authors were at CISPA Helmholtz Center for Information Security.

We first show that by selectively fine-tuning only the identified subset of layers responsible for textual content, we can significantly enhance the model's performance in generating text within images without reducing the quality and diversity of generated samples. Then, we present that by selectively applying *patching*, we are able to substitute the generated text without affecting other visual attributes of an image. Our method does not require any additional extra data (potentially with human annotations), DM training (Brooks et al., 2023), semantic maps which indicate which part of images should be preserved during the diffusion process (Andonian et al., 2021; Tuo et al., 2024), or optimization. Finally, we extend our edition technique to prevent the generation of toxic text, *on the fly* without imposing additional computational cost.

**Our contributions can be summarized as follows:**

1. We localize a small subset of cross and joint attention layers in diffusion models that determine text generated within images. Our observations are architecture-agnostic.

2. We introduce a new fine-tuning strategy that targets only the localized subset of layers responsible for textual content, improving text generation performance while maintaining the model's overall generation diversity and efficiency.

3. We incorporate our findings into the new image-to-image method for the text edition within synthetic images, outperforming previous techniques on standard benchmarks for image text editing, achieving superior accuracy and visual consistency.

4. We show that our method can also be effectively used to prevent the generation of harmful or toxic text within images in one generation pass.

## 2 BACKGROUND AND RELATED WORK

**Text-to-Image diffusion models.** Diffusion models (Song & Ermon, 2020; Ho et al., 2020) approximate data distribution by training a noise estimator $\epsilon_\theta(x_t, t, y)$ to reverse the diffusion process. The synthetic images are then generated by sampling an initial Gaussian noise, denoted as $x_T \sim \mathcal{N}(\mathbf{0}, \mathbf{I})$, and progressively removing the predicted noise at each time step $t = T, \ldots, 1$ up until obtaining clean data sample $x_0$. The noise predictor $\epsilon_\theta(x_t, t, y)$ is usually implemented as a U-Net (Ronneberger et al., 2015) or, recently, (as in SD3 (Esser et al., 2024)) a transformer-based model (Vaswani, 2017; Peebles & Xie, 2023). In common text-to-image DMs (Ramesh et al., 2022; Rombach et al., 2022; Saharia et al., 2022; StabilityAI, 2023), the conditioning input $y$ is a text embedding derived from a textual prompt $p$ using pre-trained text encoders, such as the text encoder from CLIP (Radford et al., 2021) or the large language models like T5 (Raffel et al., 2020) as used in DeepFloyd IF (StabilityAI, 2023) or SD3 (Esser et al., 2024)).

**Cross and Joint Attention layers.** The integration of text conditioning into the denoising process is achieved through cross-attention layers (Vaswani, 2017). The most standard cross-attention (used, *e.g.,*, in Stable Diffusion or SDXL (Rombach et al., 2022)) operates by computing three components: the query $Q = hW^Q$, the key $K = eW^K$, and the value $V = eW^V$, where $h$ and $e$ represent the hidden image and text representations, respectively, and $W^Q$, $W^K$, and $W^V$ are learnable weight matrices. The attention probabilities are then calculated using the following equation: $\text{Attention}(Q, K, V) = \text{softmax}\left(\frac{QK^T}{\sqrt{d}}\right) \cdot V$, where $d$ is a scaling factor equal to the dimension of the queries and keys. More recent diffusion models extend this mechanism further. Specifically, the DeepFloyd IF (StabilityAI, 2023) model implements cross-attention layers where the keys and values are formed by concatenating the projections of both $h$ and $e$. Esser et al. (2024) further advance this mechanism by introducing a so-called *joint attention*, where each attention component ($Q$, $K$, and $V$) is a concatenation of projections from both $h$ and $e$. Crucially, in this setup, both image and text projections are propagated throughout the diffusion model, in contrast to standard cross-attention layers where each attention block received the same static text-encoder embedding $e$ as input. In our work, we demonstrate that our patching technique is invariant to these implementation changes and can be applied effectively across all of them.

**Interpretability of diffusion models.** Recent works have explored the inner workings of diffusion models by analyzing cross-attention layers (Tang et al., 2023; Hertz et al., 2023). On the other hand, Park et al. (2024) explains the predictions of diffusion models at each denoising step using saliency

maps. Other research efforts have focused on localizing where specific concepts are stored within diffusion models. For instance, Hintersdorf et al. (2024) pinpoint the memorization of individual training data samples within DMs at the neuron level in cross-attention layers, using the *z-score*. Basu et al. (2024b) develop a framework utilizing causal tracing (Pearl, 2001) to identify where knowledge of various styles, objects, or facts is stored within the Stable Diffusion model (Rombach et al., 2022). In follow-up work, Basu et al. (2024a) extend this framework by introducing a mechanistic approach to knowledge localization across different text-to-image DMs. Despite being effective across models with standard cross-attention implementations, such as Stable Diffusion XL (SDXL) (Podell et al., 2024) and DeepFloyd IF StabilityAI (2023), it lacks analysis on the most recent attention variants, such as *joint attention* (Esser et al., 2024). In contrast, our approach localizes small fractions of components responsible for generating textual content and is applicable across different cross-attention variants.

**Text rendering in diffusion models.** Recent diffusion models, such as Stable Diffusion (Rombach et al., 2022), generate high-quality images conditioned on text prompts but often struggle with rendering coherent visual text. To address this limitation, more advanced DM architectures (e.g., SDXL, Deep Floyd IF, SD3 (Esser et al., 2024), and FLUX (Labs, 2024)) incorporate multiple text encoders, often based on models like CLIP (Radford et al., 2021) or large language models like T5 (Raffel et al., 2020), to enhance the quality of generated text within images.

In parallel with the above efforts, several other approaches have emerged to improve the fidelity of generated text by adding components to the generation pipeline. For example, TextDiffuser (Chen et al., 2023) employs a two-stage process where a layout transformer (Gupta et al., 2021) first identifies text coordinates as segmentation masks, which are later used to fine-tune a latent diffusion model to accurately inpaint or modify text based on prompts. Similarly, AnyText (Tuo et al., 2024) integrates an auxiliary latent module to process inputs like text glyphs or masked images and a text embedding module using OCR to blend stroke data with image caption embeddings. Additionally, other works incorporate extra conditioning during generation, such as Zhang et al. (2024b) with sketch images or Yang et al. (2024), which leverages glyph instructions.

**Fine-tuning diffusion models with LoRA.** Low-Rank Adaptation (LoRA) (Hu et al., 2022) is a fine-tuning approach known for its capacity to deliver high-quality results with both spatial and temporal efficiency. LoRA achieves this by introducing external low-rank weight matrices, which are optimized for the attention layers of the base model while keeping the pre-trained model weights unchanged. After the training process, these low-rank matrices define the adapted model, which can then be applied to the target task. Recently, (Frenkel et al., 2024) introduced B-LoRAs that leverage LoRA to explicitly disentangle the style and components of an image. In our work, we tune the localized layers using LoRA to further improve the generated text within images.

**Controlling diffusion models with cross-attention.** In Appendix A, we further describe related work on text-to-image models fine-tuning and image editing by leveraging cross-attention layers and manipulating the denoising steps through keys and values.

## 3 EXPERIMENTAL SETUP

**Benchmark.** For evaluation, we adapt two benchmarks from Yang et al. (2024) for the text editing. **SimpleBench** consists of 400 prompts following the template *'A sign that says "<keyword>".'*, while **CreativeBench** includes 400 more complex prompts adapted from GlyphDraw Ma et al. (2023), such as *'Flowers in a beautiful garden with the word "<keyword>" written.'*. The keywords used in the benchmarks are from a pool of single-word candidates from Wikipedia and categorized into four buckets based on their frequency: $\textbf{Bucket}_{\textbf{top}}^{\textbf{1k}}$, $\textbf{Bucket}_{\textbf{1k}}^{\textbf{10k}}$, $\textbf{Bucket}_{\textbf{10k}}^{\textbf{100k}}$, and $\textbf{Bucket}_{\textbf{100k}}^{\textbf{plus}}$. Both benchmarks contain the same set of keywords, which serve as text that should be generated in the images. In this work, we use 100 prompts from each benchmark, with words from $\textbf{Bucket}_{\textbf{top}}^{\textbf{1k}}$, as a *validation set*, and the remaining 300 prompts as a *test set*. The prompts from these benchmarks serve as the source prompts $p_S$. To create the target prompt $p_T$ for each $p_S$, we use the same prompt template as in $p_S$, but select the keyword from a different source prompt, ensuring that the corresponding $p_S$ and $p_T$ differ only in the keywords.

**Metrics.**  We measure two main aspects of the generations. As text alignment, we refer to the correspondence to the keyword provided in the prompt. As image alignment, we calculate the quality of the image outside of the modified text (*e.g.,* background). To measure the text alignment, we use the **OCR F1 Score**, which is calculated as follows: F1 Score $= \frac{2 \times \text{Precision} \times \text{Recall}}{\text{Precision} + \text{Recall}}$, where *Precision* measures the ratio of predicted characters in the keyword, and *Recall* measures the ratio of characters in the keyword that are covered by the prediction. Additionally, we compute the **Levenshtein distance (LD)** between the keyword and the text predicted by the OCR model and **CLIP-T Score** (Radford et al., 2021) measuring the similarity of the target text (contained in the target prompt $p_T$) and the text in the edited image. To measure the alignment between original and edited images, we calculate **Mean Squared Error (MSE)**, which is the average squared difference between the reference and generated images, indicating how close the generated image is to the reference; lower values indicate higher similarity. We also compute a **Structural Similarity Index Measure (SSIM)** (Wang et al., 2004) that evaluates the perceived quality of a generated image by comparing its luminance, contrast, and structure to a reference image, with higher values indicating greater similarity. Finally, we use the **Peak Signal-to-Noise Ratio (PSNR)**, which measures the ratio between the maximum possible power of a signal and the power of corrupting noise that affects the fidelity of its representation, where the signal, in our case, is the reference image and the noise is the error introduced by editing the image; higher PSNR values indicated greater fidelity.

**Models.**  We identify the layers responsible for text generation in the three recent DMs, namely SDXL (Podell et al., 2024), DeepFloyd IF (StabilityAI, 2023), and SD3 (Esser et al., 2024), that differ significantly in their architecture, especially in the text encoder parts and the implementations of attention layers. To detect text in generated images, we use the EasyOCR model. We choose a non-multi-modal method for this task to ensure that OCR-based metrics are computed purely based on the text present in images. We observe that multi-modal OCR models tend to guess the text based on the visual context, even when not present in the image. As a text detection model, we use the DBNet (Liao et al., 2020).

## 4 LOCALIZATION OF ATTENTION LAYERS RESPONSIBLE FOR TEXTUAL CONTENT GENERATION

We begin by presenting details of our patching technique for cross and joint attention layers, which we employ to localize the components of diffusion models responsible for the content of the generated text. We demonstrate that our method generalizes across diverse model architectures despite differences in the implementations of attention layers and with different configurations as well as types of text encoders.

### 4.1 PATCHING TECHNIQUE

Recent works (Basu et al., 2024a; Orgad et al., 2023) demonstrate that altering the key and value matrices of cross-attention layers can effectively influence the concepts generated by diffusion models. Specifically, Basu et al. (2024a) show that only certain attention layers within DMs are responsible for generating specific visual concepts, such as objects or styles. This approach that we call *injection* is effective in U-Net-based DMs such as Stable Diffusion or DeepFloyd IF, as shown in Figure 1 (**B**). These models implement cross-attention layers that directly input the prompt embedding $e$ and multiply it by the key $W^K$ and value $W^V$ matrices. However, it is unsuitable for the most recent DMs that leverage the joint attention mechanism (Esser et al., 2024), such as SD3 and FLUX. In these models, the subsequent attention layers process and modify both image and conditioning text, allowing each following layer to receive text embeddings modified by its preceding layers.

In our work, we leverage the *activation patching* technique (Meng et al., 2022) to identify the cross and joint attention layers responsible for generating text content in images across different DM's architectures. We present the overview of the patching process in Figure 1 (**A**). Suppose we want to edit the text in the image $i_S$ generated from the source prompt $p_S =$ 'A sign that says "$t_S$".' to match the text in the target prompt $p_T =$ 'A sign that says "$t_T$".'. To measure the impact of each individual cross-attention layer $l$ on the content of text generated in the output image, we first generate an image $i_T$ from $p_T$, caching the keys $K_T = e_T W_l^K$ and values $V_T = e_T W_l^V$ (**A.I**), where $e_T$ denotes the textual input part to the cross-attention layer. Then, while generating $i_S$ from $p_S$, we overwrite $K_S$ with $K_T$ and $V_S$ with $V_T$ (**A.II**). We then calculate image and text alignment

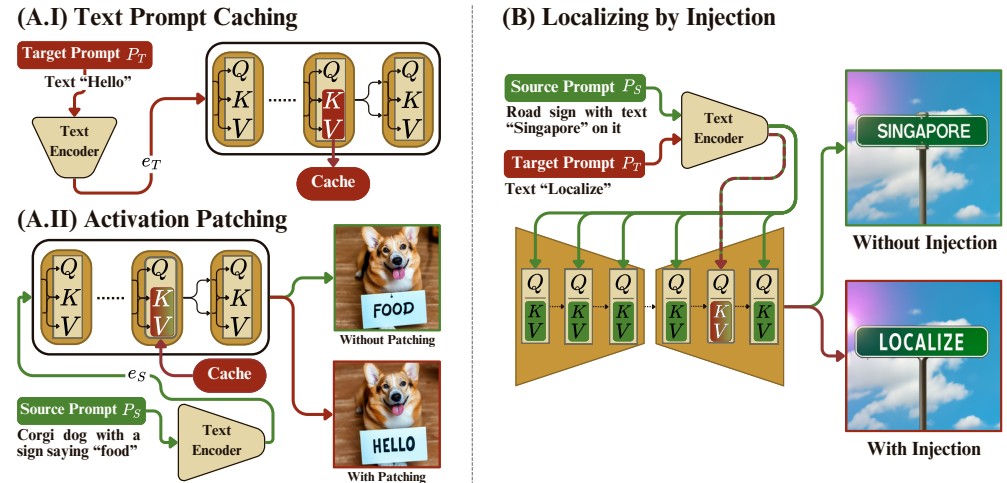

Figure 1: **Overview of the localization process.** Our goal is to edit the image generated from the source prompt $p_S$ using the target prompt $p_T$. To find which cross and joint attention layers should be modified, we pass the target prompt $p_T$ through the DM, caching the keys and values. Then, while generating the image from $p_S$ we substitute the keys and values with the cached ones. We select the layers which yield the highest image and text alignment. (A) Localizing by Patching is applied to SD3, and (B) Localizing by Injection is used for SDXL and DeepFloyd IF.

| Model | # localized layers | total # of cross--attention layers | # localized parameters | fraction of model parameters [%] |
|---|---|---|---|---|
| **SDXL** (Podell et al., 2024) | 3 | 70 | 15.7M | 0.61% |
| **DeepFloyd IF** (StabilityAI, 2023) | 1 | 22 | 8.9M | 0.21% |
| **SD3** (Esser et al., 2024) | 1 | 24 | 4.7M | 0.23% |

Table 1: **Less than 1% of DMs' parameters influence text generation within the images.**

metrics for the generations produced by the diffusion model with modified attention activations. To ensure consistency in our method, we always cache and overwrite only the *text* keys and values, which result from multiplying the textual parts of the residual stream by the key and value matrices. It allows us to apply our technique across different DM architectures despite their differences in attention implementations.'

## 4.2    CROSS-ATTENTION LAYER LOCALIZATION

We localize the layers responsible for text generation in three DMs with different architectures and text encoders: SDXL, DeepFloyd IF, and SD3. To this end, we run our patching approach for each cross-attention layer in each model on our validation set. As presented in the overview of the results in Table 1 and Figure 2, we are able to successfully identify cross-attention layers that, when patched, cause the DMs to produce the text that closely matches the text in the target prompt $p_T$. In both DeepFloyd IF and SD3 models, there is only a single layer that strongly responds when patched

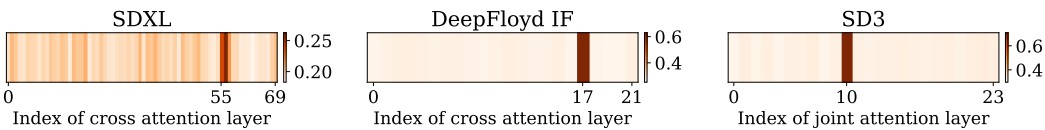

Figure 2: **Localized attention layers responsible for the content of the generated text.** We selectively patch individual cross and joint attention layers with computations for the target prompt and measure the responses with OCR F1 Score. We identify three layers with the highest responses in SDXL (55, 56, and 57), one layer in DeepFloyd IF (17), and one layer in SD3 (10).

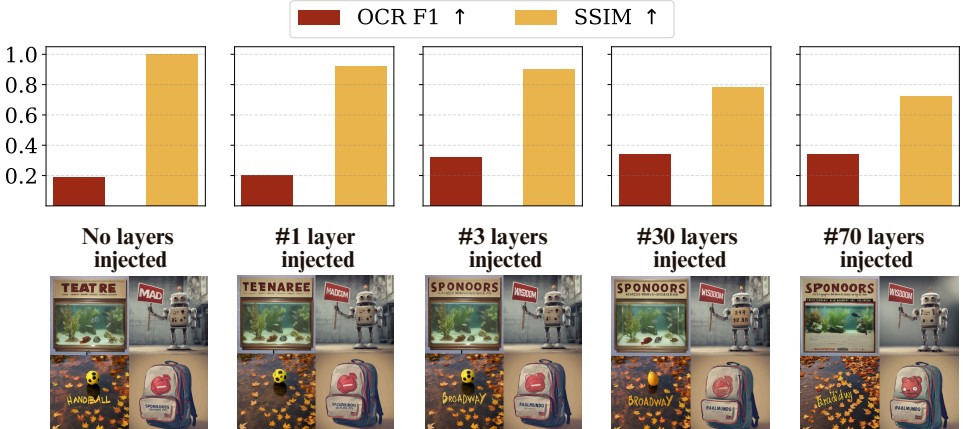

Figure 3: **The localized layers effectively balance the text alignment with the target prompt $p_T$ and the image alignment with the source prompt $p_S$.** For ease of exposition, we measure the text alignment with OCR F1 and the image alignment with SSIM. We observe that injecting the target prompt $p_T$ to too many layers decreases the image alignment and introduces undesirable artifacts, *e.g.,* the Japanese text on the robot's chest in 2nd image from the right and the lack of fish in the 1st image from the right. Conversely, injecting $p_T$ to too few layers does not edit the generated text. We present more details about the experiment in Appendix E.

| Target prompt | Model | CLIP-T | | OCR F1 | | | | |
|---|---|---|---|---|---|---|---|---|
| | | Template$_S$ | Template$_T$ | Text$_S$ | Text$_T$ | | | |
| Template$_S$:Text$_S$ | SDXL | **0.727** | 0.436 | **0.354** | 0.206 | | | |
| Template$_S$:Text$_T$ | SDXL | **0.732** | 0.436 | 0.194 | **0.324** | | | |
| Template$_T$:Text$_T$ | SDXL | **0.724** | 0.440 | 0.203 | **0.331** | | | |
| Template$_S$:Text$_S$ | DeepFloyd IF | **0.721** | 0.453 | **0.554** | 0.244 | | | |
| Template$_S$:Text$_T$ | DeepFloyd IF | **0.729** | 0.453 | 0.260 | **0.475** | | | |
| Template$_T$:Text$_T$ | DeepFloyd IF | **0.721** | 0.465 | 0.275 | **0.452** | | | |
| Template$_S$:Text$_T$ | SD3 | **0.675** | 0.443 | **0.544** | 0.231 | | | |
| Template$_S$:Text$_T$ | SD3 | **0.599** | 0.443 | 0.266 | **0.333** | | | |
| Template$_T$:Text$_T$ | SD3 | **0.684** | 0.446 | 0.276 | **0.304** | | | |

Figure 4: **Patching preserves visual components from the source prompt, taking only the textual information from the injected target prompt.** In all the combinations of templates and texts that we inject to localized layers of diffusion models (with other layers receiving both source template and source text), the final visual components of the image are always closer to the original template, while the textual content is always aligned with the one from an injected prompt. The source prompt is always defined as $p_S$=Template$_S$:Text$_S$, while we change the target prompts to Template$_S$:Text$_S$, Template$_S$:Text$_T$, and Template$_T$:Text$_T$ (from left to right for the images).

with the other prompt. On the other hand, in the SDXL model, we identify three such layers. The fact that in SDXL, the responses measured in the F1 Score are much more distributed than in other analyzed models may be attributed to the fact that SDXL has significantly more cross-attention layers than the other models and exhibits the lowest text generation capabilities. Overall, our findings suggest that a very small fraction of the DM's parameters is primarily responsible for the text content in the generated images. Additionally, the successful localization of DM components across models demonstrates the applicability of our localization method across different DM architectures. In Figure 3, we additionally visualize how patching a different number of layers affect the final generation in Stable Diffusion XL.

## 4.3 SPECIALIZATION OF THE LOCALIZED LAYERS

In the previous section, we localized layers that are responsible for the generation of the textual content. Here, we delve deeper into this analysis and evaluate their specialization. In particular, we study what is the information extracted from the prompt by the selected layers and how it affects the generation. To measure this effect, we conduct a series of experiments with artificial prompts created as a combination of a *template* that describes the background of the image and *text*, usually in the form of a simple word. We present examples of such prompts in Table 2.

We show that selected layers are only affected by the part of the target prompt that mentions the textual content. To that end, we sample images with a prompt $p_S = \text{Template}_S : \text{Text}_S$ used as conditioning for almost all the layers while patching the localized layers with one of three target prompt options: (1) the same prompt ($p_T = p_S$), (2) a prompt that shares the same template but different text ($p_T = \text{Template}_S : \text{Text}_T$) or

Table 2: Examples of prompts.

| Template | Text |
|---|---|
| *A book cover with text* | *'Love'* |
| *A sign that says* | *'STOP'* |
| *A paper letter with note* | *'Lies'* |

(3) a prompt with different template and text ($p_T = \text{Template}_T : \text{Text}_T$). We present the result of this experiment in Figure 4. We observe that the final generation follows the text provided by the prompt $p_T$ used for patching. However, at the same time, changing the template in the target prompt does not affect the final generation, as the background image is always significantly more aligned with the template from the source prompt. This observation means that the layers localized by our method are not only used for generating the textual content in the final sample but are also highly specialized, focusing solely on the textual content of the input prompt.

## 5 APPLICATIONS OF OUR METHOD

Focusing on the localization of cross and joint attention layers for text generation offers several key advantages. In this section we highlight specific use cases where it plays an instrumental role. We first show that we can precisely fine-tune selected layers to improve the quality of the generated text of a base model without affecting its remaining generative capabilities. Then, we present that with our patching technique, we can efficiently edit text from the model generations. We then extend the latter application to the cost-free technique for mitigating harmful or inappropriate text generation.

### 5.1 IMPROVING TEXT GENERATION THROUGH FINE-TUNING

We leverage our localization insights to fine-tune pre-trained DMs on the task of visual text generation. In particular, we show that by applying Low-Rank Adaptation (LoRA) only to the localized text-specific layers, we can significantly improve the quality of the generated text without affecting the model's performance on other tasks.

#### 5.1.1 TRAINING SETUP

For training, we utilize a randomly chosen subset of 74,285 images from the MARIO-LAION 10M dataset (Chen et al., 2023). In order for the training text captions to contain text that is directly presented on the corresponding training image, we construct them according to the template *'An image with text saying "<text>"'*, where "<text>" constitutes of OCR labels corresponding to the image. We compare the performance of applying LoRA to the localized layers with the baseline adaptation approach, for which we directly follow Hu et al. (2022) and apply LoRA to all cross-attention layers. We optimize both models until convergence and evaluate the quality of model generations after the next epochs on our test set introduced in Section 3.

To assess the quality of the generated text, we report OCR F1-Score and CLIP-T. Additionally, to quantify the effect of fine-tuning on the general generative capabilities of the model, we use the distribution precision and recall metrics (Kynkäänniemi et al., 2019) that measure the quality of individual samples (precision) and their diversity (recall) against the generations before fine-tuning. We adapt the original method to high-resolution generations from large diffusion models by substituting the original inception embeddings with the CLIP ones.

#### 5.1.2 FINE-TUNING RESULTS

Our results demonstrate that by fine-tuning only the three cross-attention layers, identified as instrumental for the generation of textual content, one can obtain a model yielding higher-quality visual text compared to the model with all of the cross-attention layers fine-tuned while preserving the models' generation capabilities. As presented in Figure 5 (top left), even though fine-tuning of the whole model initially converges faster towards the higher performance, after 20 epochs of training, the model starts to overfit, what can be observed as a significant drop in the recall of generated samples presented in Figure 5 (bottom left). On the other hand, when fine-tuning selected layers, we can observe steady improvement in the quality of the generated text, with little effect on the model's

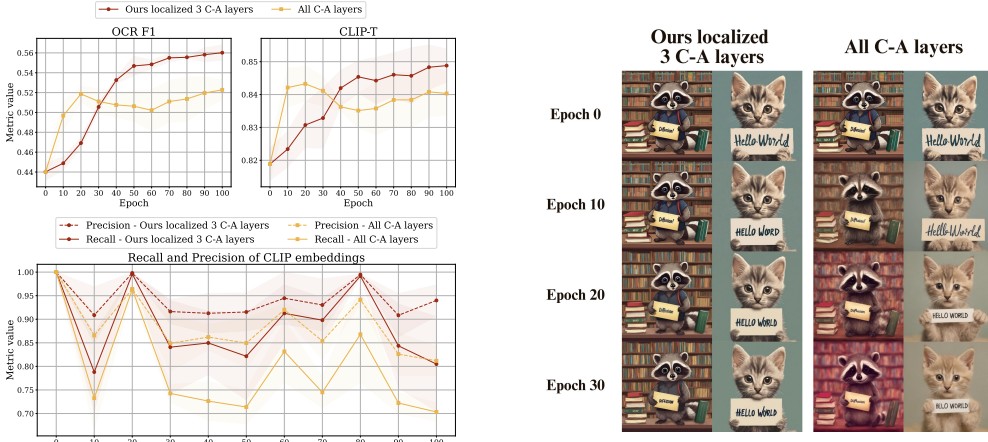

Figure 5: **Fine-tuning LoRA on localized layers improves text generation quality without compromising overall generation capabilities.** We apply LoRA fine-tuning to the SDXL model to enhance its text generation capabilities. **(top left)** The LoRA fine-tuning on the localized layers converges to a higher quality of the generated text (as measured by OCR F1 and CLIP-T metrics). **(bottom left)** When fine-tuning LoRA on all cross-attention layers (denoted as C-A), the model quickly collapses, losing its ability to generate examples that match the prompt. The diversity is significantly reduced, as indicated by a recall. In contrast, fine-tuning LoRA only on our localized cross-attention layers prevents model overfitting while improving text generation quality. It preserves diversity while achieving higher fidelity measured by precision. **(right)** We also present this effect on sample generations. Longer LoRA fine-tuning (measured in epochs) on localized layers improves text quality while preserving visual content, however, applying LoRA to all layers results in significant degradation of the image quality and diversity.

generation performance and no visible mode collapse. Additionally, Figure 5 (right) presents sample generations from different training epochs, illustrating the changes to the base model induced by fine-tuning. We focus on LoRA for SDXL since this model has a significantly lower text generation quality than other studied DMs. We also present a comparison between LoRA, the basic version of our method, and another editing method in Table 3. The results indicate that our LoRA approach outperforms the other methods on all but two metrics. Overall, it achieves superior image and text alignment while preserving the fast execution time (from the basic version of our method).

## 5.2 EDITION OF GENERATED TEXT IN IMAGES

In this section, we evaluate our patching method leveraging the localized cross-attention layers in the task of text edition on images, where the goal is to preserve most of the source prompt-driven output while selectively modifying only the regions of the image where the source and target prompts disagree. Our work can be directly compared to the prompt-to-prompt editing framework (Hertz et al., 2023) (denoted as P2P), where the image edition is controlled only by the text provided by the user. P2P also utilizes cross-attention layers in its design to modify visual concepts and defines a target prompt, which is derived from the source prompt. We evaluate both methods on SDXL, DeepFloyd IF, and SD3 models and present the results in Table 3 on our test set. Our standard patching method (denoted as "Ours") consistently outperforms P2P in terms of image alignment to the source and text alignment to the target prompt. Additionally, our approach is significantly faster in editing a single image, as reflected in the Execution Time column of the table.

While P2P is effective for DMs where the cross-attention layers' keys and values consist solely of text representations from the text encoder (such as SDXL), it struggles with models like DeepFloyd IF and SD3, where both text and image representations contribute to the keys and values. To address this, we introduce a modified version of P2P, denoted as P2P*, for these models. Instead of overwriting the entire keys and values during image generation, as in the standard approach, P2P* overwrites only the textual components of the keys, allowing image elements to change. This modification enables effective text editing according to the target prompt, albeit with more noticeable alterations to

Table 3: **Our method outperforms P2P in text editing by generating higher-quality text while preserving the other visual components.** We bold the best result for a given DM in each metric.

| Setup | Diffusion Model | SimpleBench | | | | | | CreativeBench | | | | | | Execution Time [s] ↓ |
|---|---|---|---|---|---|---|---|---|---|---|---|---|---|---|
| | | Image alignment | | | Text alignment | | | Image alignment | | | Text alignment | | | |
| | | MSE ↓ | SSIM ↑ | PSNR ↑ | OCR F1 ↑ | CLIP-T ↑ | LD ↓ | MSE ↓ | SSIM ↑ | PSNR ↑ | OCR F1 ↑ | CLIP-T ↑ | LD ↓ | |
| Ours ($t_s = 50$) | SDXL | 44.78 | 0.80 | 32.09 | 0.34 | **0.78** | 75.95 | 25.34 | 0.89 | 35.06 | 0.32 | **0.82** | 102.88 | **10.37**$_{\pm.25}$ |
| Ours ($t_s = 46$) | SDXL | 43.24 | 0.81 | 32.25 | 0.34 | **0.78** | 75.45 | 23.49 | 0.90 | 35.42 | 0.32 | **0.82** | 102.79 | **10.37**$_{\pm.25}$ |
| Ours LoRA | SDXL | **27.63** | **0.90** | **36.38** | **0.43** | 0.77 | **26.24** | **22.83** | **0.91** | **37.47** | **0.33** | 0.77 | **38.31** | **10.37**$_{\pm.25}$ |
| P2P | SDXL | 57.26 | 0.82 | 30.77 | 0.29 | 0.69 | 75.72 | 57.26 | 0.83 | 30.93 | 0.26 | 0.78 | 99.50 | 31.17$_{\pm.19}$ |
| Ours ($t_s = 50$) | DeepFloyd IF | 73.15 | 0.63 | 29.70 | **0.70** | 0.80 | 10.65 | 57.92 | 0.71 | 31.05 | 0.47 | **0.84** | 22.55 | **13.87**$_{\pm.04}$ |
| Ours ($t_s = 48$) | DeepFloyd IF | **70.27** | **0.64** | **29.90** | **0.70** | **0.81** | 10.85 | 53.50 | **0.74** | 31.46 | **0.48** | **0.84** | 21.40 | **13.87**$_{\pm.04}$ |
| P2P | DeepFloyd IF | 105.60 | 0.41 | 27.90 | 0.27 | 0.61 | **10.23** | **44.89** | **0.74** | 96.84 | 0.08 | 0.61 | **9.39** | 28.04$_{\pm.28}$ |
| P2P* | DeepFloyd IF | 105.29 | 0.21 | 27.91 | 0.41 | 0.67 | 13.48 | 44.64 | 0.67 | **96.85** | 0.11 | 0.62 | 13.80 | 28.04$_{\pm.28}$ |
| Ours ($t_s = 28$) | SD3 | 73.98 | 0.74 | 29.59 | 0.68 | 0.76 | **4.96** | 69.21 | 0.69 | 30.09 | 0.39 | 0.74 | 60.79 | **15.23**$_{\pm.19}$ |
| Ours ($t_s = 26$) | SD3 | **70.89** | 0.72 | **29.84** | 0.53 | 0.70 | 5.79 | **63.13** | 0.73 | **30.61** | 0.41 | 0.75 | 42.52 | **15.23**$_{\pm.19}$ |
| P2P | SD3 | 90.79 | **0.82** | 28.65 | 0.31 | 0.57 | 9.31 | 82.53 | **0.82** | 29.13 | 0.29 | 0.71 | 60.55 | 118.30$_{\pm.55}$ |
| P2P* | SD3 | 98.22 | 0.58 | 28.24 | **0.90** | **0.88** | **2.06** | 85.77 | 0.64 | 28.90 | **0.66** | **0.90** | 62.59 | 118.30$_{\pm.55}$ |

the source image. Furthermore, in our visual text modification approach, the target prompt $p_T$ can differ from the source prompt $p_S$ in the prompt length and positions of tokens representing the text to change, as opposed to the P2P approach. In the Appendix K, we present example edition results for our localization-based text edition method. In particular, we show that we can modify texts of varying lengths with our method.

## 5.3 Preventing generation of toxic text

We observe that diffusion models, even the ones equipped with safeguards against generating NSFW (Not Safe For Work) content, tend to simply copy-paste the text from the prompt to the image. As a result, while the visual content may be safe thanks to careful filtering of the fine-tuning dataset, the text in the generated images can still be harmful. We carry out experiments on known methods, such as Safe Diffusion (Schramowski et al., 2023) and Negative Prompts (Max Woolf, 2022), to evaluate their effectiveness in preventing the generation of toxic content and find out that those methods underperform. To address this issue, we propose a new approach – the application of our edition technique to prevent the generation of toxic text within images.

Our goal is to address scenarios where a model provider exposes a diffusion model for generating images from textual prompts. In this setting, a user may submit a source prompt $p_S$ containing toxic textual content intended to appear in the generated image. Detecting toxicity in the images is crucial for online platforms to enforce community guidelines and remove inappropriate material. With advancements in large language models, toxic text can be reliably identified (Zhang et al., 2024a) and rephrased to ensure that the generated image suppresses harmful content. To achieve this, the toxic portion of the source prompt is replaced with a non-harmful text or a placeholder sequence, such as a series of stars (*).

We harness our precise localization of the cross-attention layers responsible for generating textual content in images to prevent the model from outputting harmful text. In particular, the prompts identified as toxic are substituted with a non-harmful text *on the fly* using our patching technique. This allows us to remove the toxic content from the final generation without altering the remaining visual content. We achieve this result with a single pass through the diffusion denoising process without imposing any additional computational cost.

In Table 4, we compare our method with three baseline techniques. First, we leverage negative prompting. It was suggested (Max Woolf, 2022) that the generative process can be more effectively guided by using *negative* text prompts that instruct a diffusion model to exclude specific elements from its generated images. In that approach, we set the negative prompt to *'text "<word>"'*, where <word> is a harmful word from $p_S$. We also run Safe Diffusion (Schramowski et al., 2023) on safe prompts, which works by intervening directly in the latent space of diffusion models to remove and suppress inappropriate content during image generation. Additionally, we introduce Safe Diffusion*, where we adapt the method (its safe prompts) to the task of toxic language removal. We present the details of adaptation in Appendix F.

Table 4: **Our method can be used to prevent the generation of toxic text in images.** We bold the best result for a given DM in each metric and the runner-up is underlined.

| Method | Diffusion Model | MSE ↓ | SSIM ↑ | PSNR ↑ | OCR F1 ↓ | Toxicity score ↓ |
|---|---|---|---|---|---|---|
| Ours | SDXL | **48.20** | 0.79 | 31.68 | 0.20 | 0.003 |
| Negative prompt | SDXL | 77.95 | 0.71 | **31.76** | 0.23 | 0.052 |
| Safe Diffusion | SDXL | 49.46 | **0.81** | 31.33 | 0.34 | 0.222 |
| Safe Diffusion* | SDXL | 49.41 | **0.81** | 31.33 | 0.33 | 0.209 |
| Prompt Swap | SDXL | 79.41 | 0.66 | 31.65 | **0.19** | **0.000** |
| Ours | DeepFloyd IF | 74.96 | 0.61 | 29.60 | 0.32 | 0.018 |
| Negative prompt | DeepFloyd IF | 100.50 | 0.37 | 28.12 | 0.59 | 0.250 |
| Safe Diffusion | DeepFloyd IF | 64.30 | 0.73 | 30.19 | 0.79 | 0.555 |
| Safe Diffusion* | DeepFloyd IF | **63.65** | **0.74** | **30.25** | 0.79 | 0.540 |
| Prompt Swap | DeepFloyd IF | 100.99 | 0.35 | 28.10 | **0.30** | **0.015** |
| Ours | SD3 | 72.61 | 0.70 | 29.72 | 0.32 | 0.018 |
| Negative prompt | SD3 | 101.63 | 0.53 | 28.08 | 0.77 | 0.407 |
| Safe Diffusion | SD3 | 34.99 | 0.86 | 34.25 | 0.73 | 0.571 |
| Safe Diffusion* | SD3 | **33.67** | **0.87** | **34.56** | 0.73 | 0.568 |
| Prompt Swap | SD3 | 98.58 | 0.51 | 28.22 | **0.30** | **0.015** |

In our approach, we replace the toxic word in the source prompt $p_S$ with a non-harmful suggestion and form the target prompt $p_T$. We also include a potential method, that, similarly to us, is based on prompt edition, which we refer to as Prompt Swap. In this method, we apply the LLM-rephrased non-toxic prompt to the entire diffusion model instead of doing it only for our localized layers.

In the experiments, each of the source prompts $p_S$ (we use 400 in total) contains a harmful word from LDNOOBW (2020). We obtain the edited generations from each approach, run the OCR on the output images, and for the text returned from OCR, we calculate the toxicity score using the RoBERTa-based classifier (Liu et al., 2022). We show that Negative Prompt and Safe Diffusion (in both versions) methods are incapable of removing toxic textual content from generated images. For Prompt Swap, we observe that this method marginally outperforms our approach in toxic text prevention. However, the introduced change in the modified prompt strongly impacts other visual aspects of an image, which is not the case for our solution.

In the Appendix G, we argue that preventing the change of visual attributes, even when the end user did not see the original image, is important in order to, i.e., preserve the emotions expressed in the original prompt to the model. We demonstrate that our approach successfully substitutes toxic text from the generated images without significantly altering the remaining part of the generation, making it the most reliable solution. We include example generations and detailed evaluation supporting this claim in Figure 12.

## 6 CONCLUSIONS

This work identifies critical cross and joint attention layers in diffusion models that directly influence text generation within images. Our proposed patching method is adaptable to various diffusion model architectures, regardless of the text encoder used. We demonstrate that in SDXL, only three layers (out of 70) impact text generation, while in DeepFloyd IF and SD3, only a single layer is responsible for the generated text (out of 22 and 24, respectively). Fine-tuning these localized layers using LoRA significantly improves the quality of the generated text of a base model without affecting its remaining generative capabilities. This selective targeting approach also increases the efficiency and precision of image-editing methods applied to text, reducing unintended modifications to non-textual visual elements. Additionally, our method can be leveraged to create an effective safeguard against the generation of harmful or toxic text in images, further highlighting its practical utility in safer and more efficient text-to-image generation workflows.

## ACKNOWLEDGMENTS

The project was funded by German Research Foundation (DFG) within the framework of the Weave Programme under the project titled "Protecting Creativity: On the Way to Safe Generative Models", funding number 545047250. The project was also supported by the National Science Centre, Poland, grants no: 2023/51/B/ST6/03004 and 2022/45/B/ST6/02817, PLGrid grant no. PLG/2024/017266, and by the Warsaw University of Technology within the (IDUB) programme.

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

# APPENDIX

## A  RELATED WORK ON MANIPULATING DIFFUSION MODELS WITH CROSS-ATTENTION

Recent works introduced methods that leverage cross-attention layers for better control of text-to-image models. In Kumari et al. (2023), the authors present an efficient way of customization of text-to-image diffusion models by fine-tuning a subset of cross-attention layer parameters. While their approach demonstrates that targeting the key and value matrices in all the cross-attention layers is sufficient to introduce new concepts, we reveal that fine-tuning those matrices in fewer than 5% of cross-attention layers (see Table 1) enables better quality of the generated text.

Geyer et al. (2024) presents a framework that enables video editing using text-to-image diffusion models. Specifically, the authors introduce a method of editing the keyframes by extension of self-attention layers in which the keys from all timeframes are concatenated in order to encourage the frames to share a global appearance. The presented solution offers an effective approach to the semantic video edition.

Prompt-Mixing (Patashnik et al., 2023) enables users to explore different shapes of objects in an image. In order for objects to stay in the same positions but change their appearance, the method operates in the inference time and, in different denoising timestep intervals, injects different prompts into the cross-attention layers. In our work, we use a similar injection mechanism that we apply only to the selected text-controlling layers. We evaluate the effect of injection at different denoising steps in the Figure 6.

Cross-Attention Refocusing (Phung et al., 2023) is a calibration technique enabling better attending of tokens representing objects to image regions. By performing multiple intermediate latent optimizations by using CAR loss and Self-Attention Refocusing loss, authors achieve better controllability of the layout of generated objects. Similar to our work, CAR focuses on cross-attention maps but aims to strengthen attention to the correct token while reducing it elsewhere.

Plug-and-Play (Tumanyan et al., 2023) is an effective image-to-image translation method. In this work, the authors show that in the denoising procedure, one can extract spatial features from the U-Net decoder's Residual Blocks and their following self-attention layers, obtaining encodings of the composition of the image. Next, by passing a different prompt during the denoising procedure for the same initial Gaussian noise, one can inject previously extracted features and obtain generations, differing in image attributes specified in the condition. In this work, we show that by focusing on text-related features we can perform a precise edition by targeting a single attention layer.

## B  SELECTION OF DENOISING TIMESTEPS

To further refine the identification of text generation capabilities in DMs, we investigate from which point in the diffusion denoising process the key and value matrices should be patched to achieve the highest performance in text editing. We present the results of this analysis in Figure 6. We observe that when starting the patching from the later timesteps $t$, we can observe better preservation in the visual attributes of a modified image and improve the quality of the generated text, increasing its similarity to the text from the target prompt $p_B$. This trend aligns with the work by Hertz et al. (2023), where authors show that only the overall structure of an image is generated in the initial steps of the diffusion denoising process. Thus, in order to reduce the change in visual attributes, we apply our patching method to localized attention layers starting from timesteps: $t_s = 46$ for SDXL, $t_s = 26$ for SD3, and $t_s = 48$ for DeepFloyd IF. Attention activations from timestep $T$ to $t_s - 1$ remain unchanged while we patch all activations from timestep $t_s$ to 0.

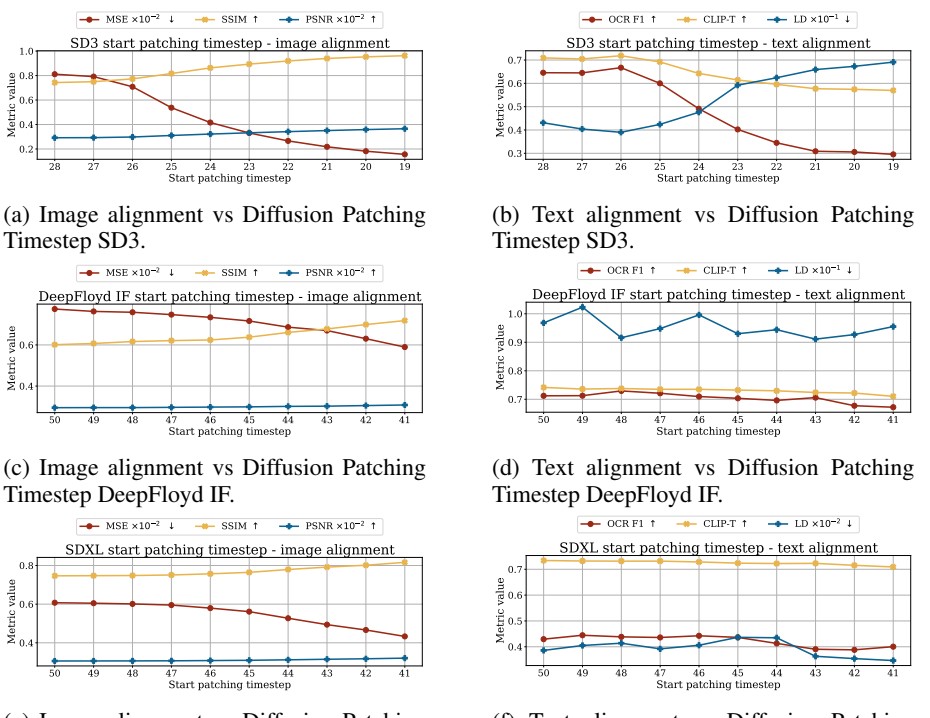

(a) Image alignment vs Diffusion Patching Timestep SD3.

(b) Text alignment vs Diffusion Patching Timestep SD3.

(c) Image alignment vs Diffusion Patching Timestep DeepFloyd IF.

(d) Text alignment vs Diffusion Patching Timestep DeepFloyd IF.

(e) Image alignment vs Diffusion Patching Timestep SDXL.

(f) Text alignment vs Diffusion Patching Timestep SDXL.

Figure 6: **Starting the text edition from a later diffusion timestep improves both image and text alignment.** We analyze the impact of the diffusion timestep from which we start the patching on the image and text alignment. We observe that we can find an optimum diffusion timestep that can simultaneously improve image and text quality.

## C  LoRA FINE-TUNING ACROSS DIFFERENT SETUPS

To further strengthen the evidence that we have correctly identified the cross-attention layers responsible for the content of the generated text, we conduct the LoRA fine-tuning process on other sets of three cross-attention layers. These sets are selected based on the OCR F1 Scores presented in Figure 2 — specifically, we select three sets of adjacent layers with the *highest* and *lowest* sum of F1 scores, respectively. As shown in Figure 7, we observe a significant performance gap between the fine-tuned layers we localized and any other set of layers. Notably, some of the chosen layer sets even decrease performance compared to the base SDXL model.

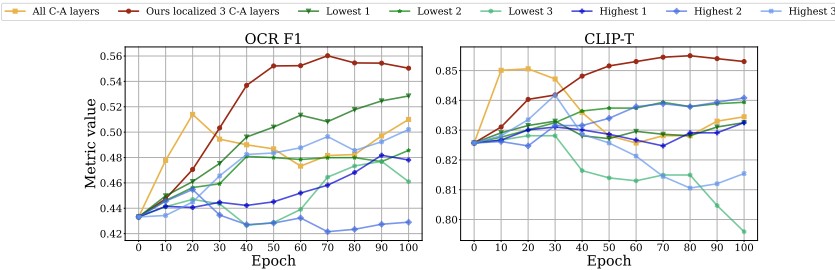

Figure 7: **LoRA SDXL Fine-Tuning Across Different Setups.** We fine-tune LoRA applied to the SDXL model to improve the text generation capabilities of the base model. When we fine-tune LoRA on all cross-attention layers, the model quickly collapses and loses its ability to generate examples that match the prompt. In contrast, when we fine-tune LoRA only on our localized three cross-attention layers, we successfully prevent model overfitting while also improving text generation quality. This trend is not observed when we apply LoRA to other sets of three layers.

# D  LoRA Fine-tuning with different training set sizes

To evaluate how our findings from Section 5.1 generalize to varying training set sizes, we fine-tune LoRA applied to the SDXL model on datasets ranging from 20k to 200k samples. To mitigate potential overfitting, especially in configurations where LoRA is applied to every cross-attention layer (*Full model* setup), we scale the training set size up to 200k samples. We train each setup for 12k steps with a batch size equal to 512 and a learning rate of 1e-6.

In Figure 8, we plot the recall and precision metrics across training steps. Notably, even with a substantially larger dataset in the *Full model 200k* configuration, the model exhibits a similar collapse to what is observed when training on smaller subsets. Moreover, both recall and precision remain largely unchanged across different setups, demonstrating the robustness of our approach, which focuses on fine-tuning specific layers.

Additionally, in Figure 9, we plot the OCR F1 Score and CLIP-T metrics, highlighting that fine-tuning localized layers, even with as few as 20k samples, results in better performance than the *Full model* setup trained with 200k samples.

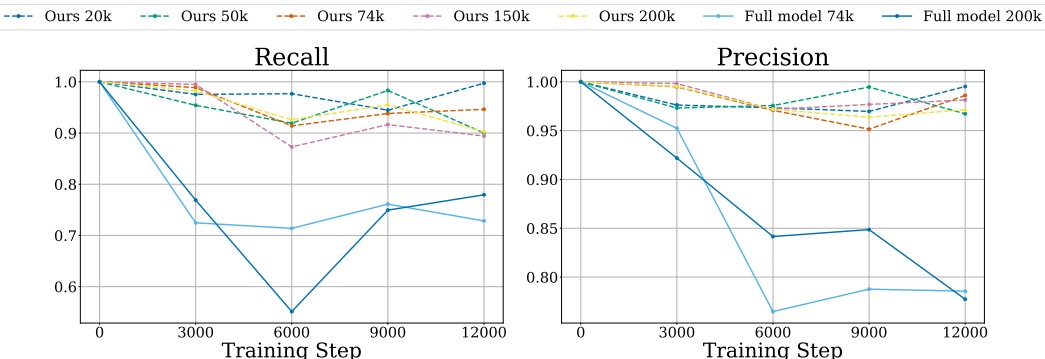

Figure 8: **Scaling up training size when fine-tuning all cross-attention layers does not prevent model collapse.** Increasing the training dataset size fails to mitigate model collapse, as evidenced by the significant drop in Recall and Precision metrics. In contrast, our approach, which fine-tunes only localized cross-attention layers, demonstrates consistent performance regardless of training set size.

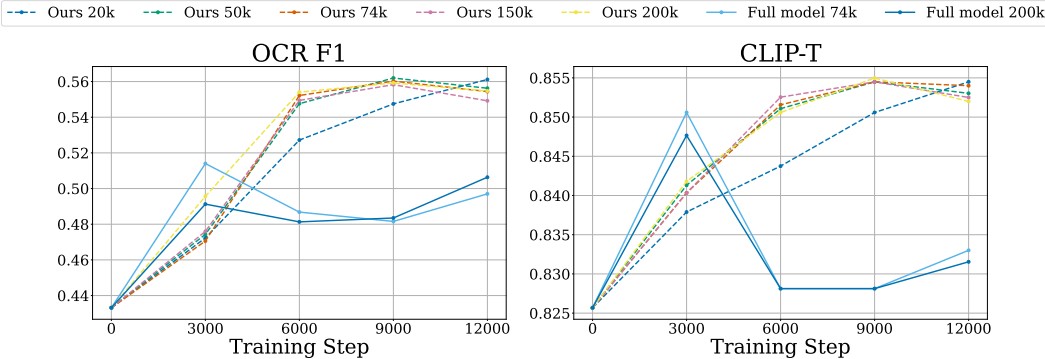

Figure 9: **LoRA fine-tuning of localized layers outperforms fine-tuning of all cross-attention layers, even with smaller datasets.** LoRA fine-tuning of localized layers achieves consistent performance across all evaluated training set sizes, from 20k to 200k samples. While increasing the dataset size slightly improves the performance of the model when all cross-attention layers are fine-tuned, a noticeable performance gap remains compared to localized fine-tuning.

# E    STUDY ON THE NUMBER OF INJECTED LAYERS IN SDXL

We carry out the study on the number of injected layers in SDXL in Table 5. We observe that leveraging more layers for the injection implies a higher alignment of visual text to the target prompt while lowering the background preservation to the source prompt. Using 3 layers in the Stable Diffusion XL model leads to obtaining the final image with text nearly as good as if injected to all the layers, yet preserves the background close to 1-layer injection.

Table 5: **Preservation-edition trade-off in SD-XL**. Injecting the target prompt into more layers enhances the text edition but also preserves less background from the source prompt.

| # layers injected | Image Alignment | | | OCR F1 | | CLIP-T | |
|---|---|---|---|---|---|---|---|
| (layers idx) | MSE $\downarrow$ | SSIM $\uparrow$ | PSNR $\uparrow$ | Text$_S \downarrow$ | Text$_T \uparrow$ | $p_S$ | $p_T$ |
| **0** (-) | 0.00 | 1.00 | 148.13 | 0.34 | 0.19 | 0.85 | 0.71 |
| **1** (55) | 17.63 | 0.92 | 36.88 | 0.28 | 0.20 | 0.82 | 0.73 |
| **2** (55,56) | 22.27 | 0.90 | 35.73 | 0.20 | 0.30 | 0.75 | 0.81 |
| **3** (55,56,57) | 23.38 | 0.90 | 35.43 | 0.19 | 0.32 | 0.74 | 0.82 |
| **10** (54,55,...,63) | 25.02 | 0.89 | 34.97 | 0.19 | 0.33 | 0.74 | 0.82 |
| **30** (34,36,...,63) | 53.83 | 0.78 | 31.13 | 0.19 | 0.34 | 0.73 | 0.84 |
| **70** (0,1,...,69) | 65.98 | 0.72 | 30.17 | 0.19 | 0.34 | 0.72 | 0.85 |

For the images presented in Figure 3, we use the following prompts.

Source prompts:

```
['A sign saying "teatre" in the aquarium',
 '"handball" written on autumn leaves floating on the lake',
 'A robot is holding a sign that says "mad"',
 'The slogan "sponsors" is written on the schoolbag']
```

Target prompts:

```
['A sign saying "sponsors" in the aquarium',
 '"broadway" written on autumn leaves floating on the lake',
 'A robot is holding a sign that says "wisdom"',
 'The slogan "raimundo" is written on the schoolbag']
```

# F    SAFE DIFFUSION SAFETY CONCEPTS

To reproduce the results for Safe Diffusion (Schramowski et al., 2023) in Section 5.3 we first run this method to all of analyzed models on the original safety concept introduced in the Safe Diffusion work: *"hate, harassment, violence, suffering, humiliation, harm, suicide, sexual, nudity, bodily fluids, blood, obscene gestures, illegal activity, drug use, theft, vandalism, weapons, child abuse, brutality, cruelty"*. Additionally, for a fair comparison, we also ask ChatGPT to extend this safety concept in order to prevent generation of toxic textual content on an image to: *"an image showing hate, harassment, violence, suffering, humiliation, harm, suicide, sexual, nudity, bodily fluids, blood, obscene gestures, illegal activity, drug use, theft, vandalism, weapons, child abuse, brutality, cruelty, profanity, toxic language, slurs, derogatory remarks, demeaning or offensive language, racist, sexist, homophobic, xenophobic, or otherwise harmful content aimed at individuals or groups, hate speech, inflammatory remarks, insults, and any form of dehumanization"*.

## G    ON PREVENTING TOXIC GENERATIONS WITH PROMPT SUBSTITUTION

The simplest solution for preventing toxic textual content generation with text-to-image models is automatically detecting and substituting the harmful part of the original user's prompt. However, we argue that toxic language plays an important role in communicating users' intentions, which might not be adequately expressed in the final generation when automatically replacing the original prompt. As observed by Jay & Janschewitz (2008) *„The main purpose of swearing is to express emotions, especially anger, and frustration"*, so even though we might not want to directly generate samples with toxic text, we still want them to align with user's emotions, and reflect them in the generation. Nevertheless, because of the cross-attention mechanism, where tokens representing text can influence other aspects of the image, this might not be possible with automatic prompt replacement. To further highlight this problem, we run a simple experiment where we measure the influence of textual content on the emotional tone of the final generation. To that end, we generate samples with Stable Diffusion 3 model, following the prompt:

*„A photo of a young man holding a sign „[XYZ]", visible strong emotion on his face"*

where we replace [XYZ] with either toxic text, or it's clean version as suggested by ChatGPT as presented in Table 6.

Table 6: Examples of toxic prompts with their substitutions as suggested by ChatGPT when asked to remove only toxic words with as small changes to the text as possible. (∗ added by authors).

| Toxic text | LLM-suggested replacement |
|---|---|
| *F∗∗k this s∗∗t* | *Forget this mess* |
| *Go f∗∗k yourself, d∗∗k* | *Go fix yourself, dude* |
| *F∗∗k off, i∗∗ot* | *You're a pain* |
| *You d∗∗b f∗∗k* | *Back off, fool* |
| *S∗ ∗ ∗w you j∗∗k* | *S∗ ∗ ∗w you, pal* |
| *You f∗∗king m∗ ∗ ∗n* | *You clumsy m∗ ∗ ∗n* |
| *You're so f∗∗ked* | *You're so doomed* |

We sample 10 generations for each prompt and their replacement suggestions, with the same random seeds pairwise. Given a set of toxic and non-toxic generations, we compare the expression we read from the face of a man holding a toxic or non-toxic sign. To that end, we use the DeepFace library (Serengil & Ozpinar, 2021) that implements methods for detecting seven basic emotions from facial expressions. In Figure 10, we present the results of this comparison. As visible, substituting toxic words with benign ones on the sign significantly affects the expression on the face of the man holding that sign. This can be especially visible with the reduced score for *angry* and higher score for *neutral* expressions. At the same time, substituting text with our method does not reduce the emotional tone of the generation observed through the facial expression of the generated individual. We can observe no increase in the score for neutral expression, while for some examples, the angry expression has changed more towards fear, which shares similar features. We present several generations from this experiment in Figure 11.

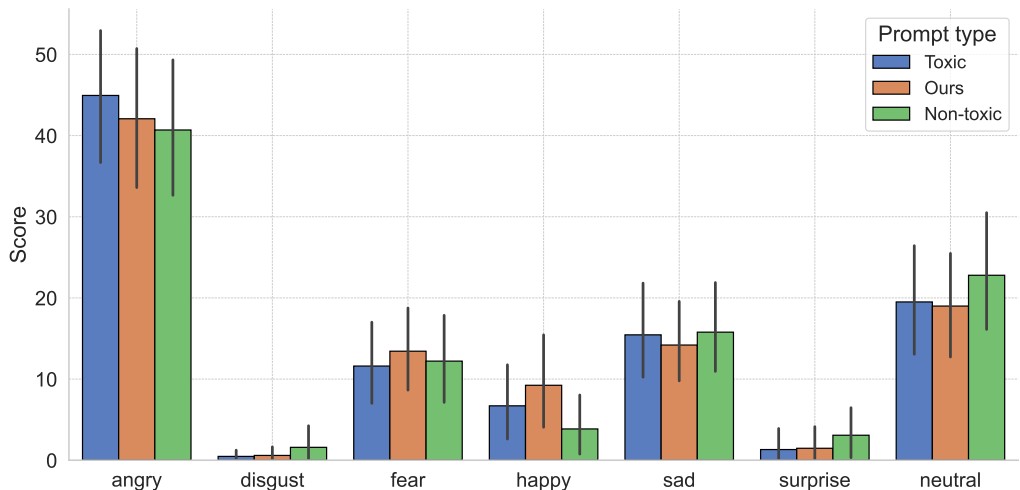

Figure 10: Comparison of facial expression scores (average), extracted from generations of a man holding a sign with toxic texts. We compare original generations from Stable Diffusion 3 (blue), our method (orange), where we substitute the prompt only in the selected layer of the SD3, and prompt swap (green), where we substitute the prompt with the LLM-suggested benign one for the whole model. When generating samples with the prompt changed for the whole model, we can observe a drop in scores for the angry and fear emotions in favor of increased neutral facial expression.

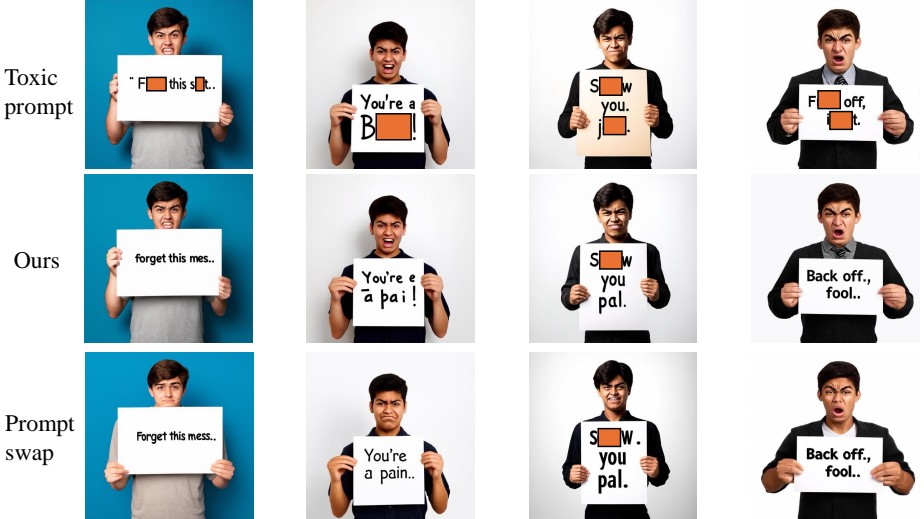

Figure 11: Influence of generated text on the final generation. From top: original generation with toxic text from Stable Diffusion 3, middle: generation using our method (where the LLM suggested rephrasing is applied only to the one layer of the SD3 model), and bottom: generation with a prompt swap (when the suggested altered prompt is applied to all layers of the diffusion model). **Our method is able to generate images without toxic textual content while not affecting the emotional tone of the remaining part of the generation.**

## H    TOXIC TEXT PREVENTION EXAMPLES

In Figure 12, we show, for the Deepfloyd IF model, the qualitative comparisons of our method to Negative Prompt, Safe Diffusion, and Prompt Swap.

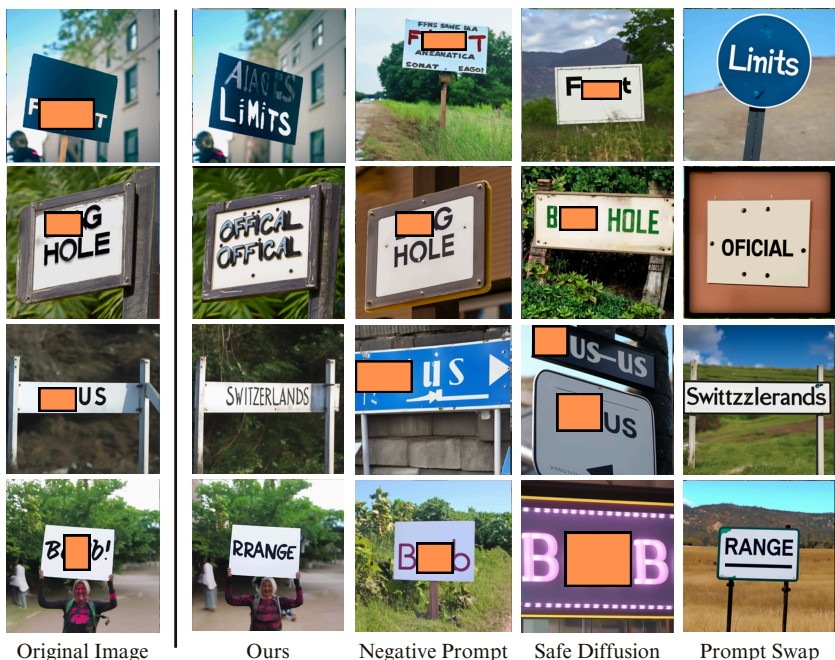

| Original Image | Ours | Negative Prompt | Safe Diffusion | Prompt Swap |

Figure 12: Example results for methods for preventing toxic text in generated images. Negative Prompt and Safe Diffusion methods are incapable of removing foul words from the images. In Prompt Swap, the background of generated images is highly influenced by the suggested word. **We show that our method successfully changes foul words yet ensures minimal changes to the other visual aspects of the image.** Orange bounding boxes were added by the authors to cover four words.

## I    PSEUDOCODE FOR LAYER LOCALIZATION

We present in Algorithm 1 our method for creating a subset of diffusion model layers that control the content of visual text generated on images.

---

**Algorithm 1** Finding subset of layers $L_{ours}$ responsible for textual content generation

---

**Require:** $P_S$: set of source prompts, $P_T$: set of target prompts, $L$: set of indices of cross-/joint-attention layers, $\theta$: threshold for acceptable OCR $F_1$-Score difference

**Ensure:** $L_{ours}$: set of selected cross-attention layers

$\quad L_{F_1} \leftarrow [\,]$ $\qquad\qquad\qquad\qquad\qquad\qquad\qquad$ ▷ initialize list of mean $F_1$-Scores for layers

$\quad L_{ours} \leftarrow \emptyset$

$\quad N \leftarrow |P_S|$

$\quad$**for** $l \in L$ **do** $\qquad\qquad\qquad\qquad\qquad$ ▷ compute $F_1$-Scores for each layer via patching

$\quad\quad I_{1..N} \leftarrow$ images generated with $L \setminus \{l\}$ receiving $P_S$ and $l$ receiving $P_T$

$\quad\quad T_{1..N} \leftarrow$ text detected in $I_{1..N}$ using an OCR model

$\quad\quad S_{1..N} \leftarrow F_1$-Score between $T_{1..N}$ and $P_T$

$\quad\quad L_{F_1}[l] \leftarrow \frac{1}{N}\Sigma_{1..N} S[i]$

$\quad$**end for**

$\quad l_{max} \leftarrow \arg\max_l L_{F_1}$

$\quad L_{ours} \leftarrow \{l_{max}\}$

$\quad$**for** $l \in L \setminus \{l_{max}\}$ **do** $\qquad\qquad\qquad\qquad\qquad$ ▷ create a set of text control layers

$\quad\quad$**if** $(L_{F_1}[l_{max}] - L_{F_1}[l]) < \theta$ **then**

$\quad\quad\quad L_{ours} \leftarrow L_{ours} \cup \{l\}$

$\quad\quad$**end if**

$\quad$**end for**

$\quad$**return:** $L_{ours}$

---

## J PARAMETER LOCALIZATION FOR THE TEXT STYLE

In this section, we examine whether the cross-attention layers we localize in Section 4.2 control not only the content of the visual text generated in the images but also its style.

**Experiment setup.** We use the Stable Diffusion 3 model, which, of all those tested, exhibits the best accuracy in generating text with the style specified in the prompt. We target four text styles: **handwritten**, **neon**, **graffiti** and **comic**. In this setup, both our source prompts $p_S$ and target prompts $p_T$ contain the same textual content to be generated but differ in the style of the text. In our experiments, we generate four sentences with the diffusion model: *'hello world!'*, *'happy new year'*, *'I love you'*, and *'Welcome to Asia'*. To ensure generalization and make sure that we do not localize layers for individual prompts, we use four prompt templates:

```
['Road sign with a {style} text saying {sentence}',
'Notebook page with a {style} text saying {sentence}',
'Street wall covered in {style} text saying {sentence}',
'Bus stop advertisement with {style} text saying {sentence}',
'Urban skatepark ramp with {style} text saying {sentence}']
```

For measuring how a particular layer $l$ controls the style of the text, we perform the patching technique in the same way as described in Section 4.1 and calculate CLIP-T alignment between the generated images (after patching the keys and values in joint-attention layer $l$) and texts *'text in s style'* where $s$ is a style from a target prompt $p_T$.

**Results.** In Figure 13, we show that the layer we localize in Section 4.2 for controlling content in visual text generated does not control the style of the text (left). Furthermore, we show (right) that in the Stable Diffusion 3 model, there is no single layer indicating the style of the generated text and that control over style in this model is distributed over multiple layers. To support this claim, we perform a study where we iteratively add the next layers with the highest response in the previous experiment and check how many of them are needed for the style to be modified. As shown in Figure 14, it is necessary to patch at least 7 out of 24 layers to change the style of the generated text. However, the images resulting from patching so many layers are also significantly different in terms of other visual aspects. This shows that there is no layer-based separation of text style from the rest of the image elements in the Stable Diffusion 3 model, which makes our observations regarding textual content even more unique.

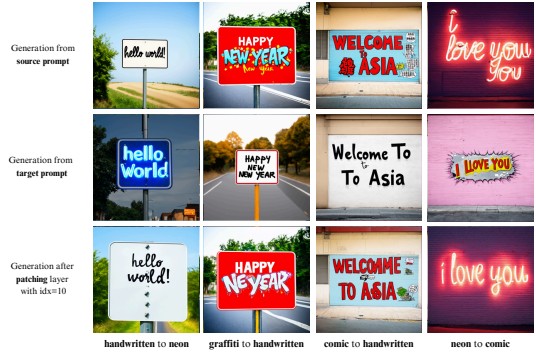 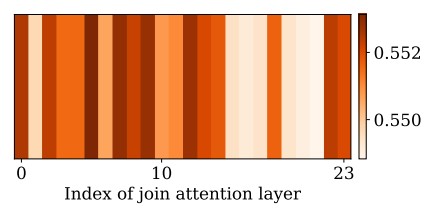

Figure 13: **The text style is not controlled by the same layer as the textual content.** We show example generations (left) indicating that the layer we localize for determining the content of the text in generated images is not capable of changing the style of the text in images. Also, we show (right) that control over the style of the text is distributed over multiple cross-attention layers in SD3 by plotting and calculating CLIP-T alignment between generations after patching particular layers with the desired text style.

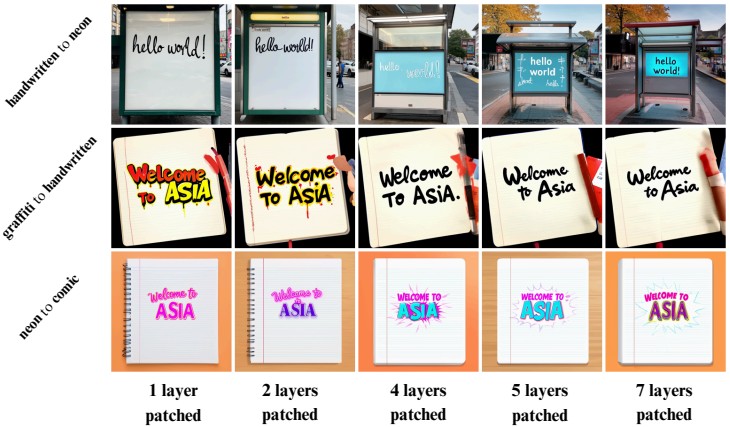

Figure 14: **The style of the text in Stable Diffusion 3 is influenced by at least 7 layers.** We provide results demonstrating performance in editing textual style when patching an increasing number of layers in the diffusion model. Although modifying this feature becomes feasible with 7 layers, it significantly alters the image background as well.

## K    IMAGE EDITION ON LONGER TEXTS

In the Figure 15, we include examples of text editing realized using our method for DeepFloyd IF (a) and Stable Diffusion 3 (b) models. Presented generations indicate that our localization technique can be used to edit images with a longer visual text. Some examples contain errors like omitted letters or words. We believe that our performance in text-based image editing strongly relies on the quality of the text generated by the diffusion model.

Additionally, we present text and image alignment metrics for image edition with our approach for varying number of words in the prompt in Table 7.

Table 7: **Performance metrics of SD3 image edition for varying number of words.**

| # words | MSE ↓ | SSIM ↑ | PSNR ↑ | OCR F1 ↑ | CLIP-T ↑ |
|---|---|---|---|---|---|
| 1 | 0.677 | 0.695 | 0.302 | 0.377 | 0.746 |
| 2 | 0.706 | 0.675 | 0.300 | 0.403 | 0.717 |
| 3 | 0.703 | 0.676 | 0.300 | 0.442 | 0.721 |
| 4 | 0.725 | 0.668 | 0.298 | 0.457 | 0.714 |
| 5 | 0.726 | 0.664 | 0.298 | 0.474 | 0.698 |
| 6 | 0.718 | 0.663 | 0.299 | 0.487 | 0.701 |
| 7 | 0.724 | 0.654 | 0.298 | 0.489 | 0.704 |
| 8 | 0.735 | 0.653 | 0.297 | 0.494 | 0.695 |

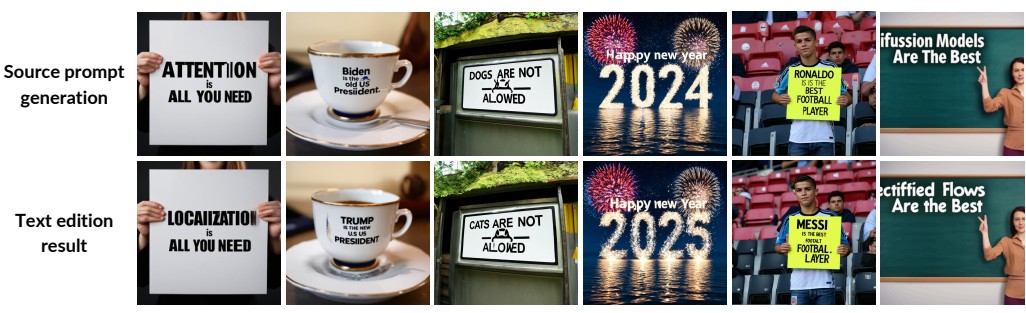

(a) DeepFloyd IF

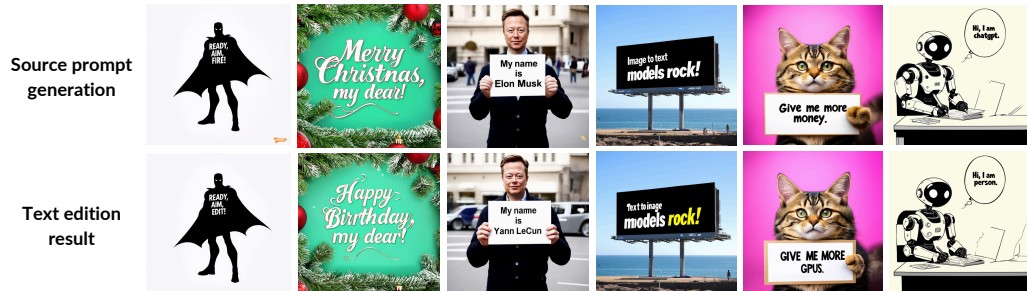

(b) Stable Diffusion 3

Figure 15: **Example results from editing synthetic images by leveraging parameter localization.**
Presented generations show that the edition can be performed for images with varying lengths of text.
We show generations for models capable of generating longer visual texts: DeepFloyd IF (a) and
Stable Diffusion 3 (b).

