# OpenReview forum: "Precise Parameter Localization for Textual Generation in Diffusion Models"
_ICLR.cc/2025/Conference — ICLR 2025 Poster_

### Official Review · Reviewer_mH7c · 2024-10-28

**Soundness:** 3
**Presentation:** 3
**Contribution:** 3
**Rating:** 6
**Confidence:** 4

**Summary:**

The paper introduces a method for precisely localizing the parameters within diffusion models (DMs) responsible for generating textual content in images. The key findings reveal that less than 1% of parameters, specifically in cross-attention and joint-attention layers, influence text generation. This localization approach, applicable across various DM architectures, including U-Net and transformer-based models, enables several enhancements:
1. Efficient finetuning: By targeting only the localized attention layers, fine-tuning improves text generation capabilities while preserving image quality and diversity.
2. Text editing: The method allows for precise text editing within generated images without altering other visual attributes.
3. Toxic content prevention: The localized layer approach provides a cost-free way to prevent generating harmful text in images.
The authors validate their method across different models, demonstrating improvements in text alignment and visual consistency while showcasing its adaptability and efficiency across architectures.

**Strengths:**

The paper is innovative in isolating specific cross and joint attention layers that directly influence text generation in diffusion models. This novel approach to localizing model parameters is broadly applicable across various architectures, including U-Net and transformer-based models, making it an impactful step forward in the interpretability and control of DMs. Furthermore, applying this localization technique to prevent toxic text generation is a unique and practical application, expanding the scope of ethical safeguards in generative models.
This research has notable significance for both practical applications and theoretical advancements in generative AI. By pinpointing and leveraging specific layers responsible for text content, the paper introduces a method that improves the efficiency and quality of DMs in text generation tasks. The application to toxic text prevention adds a meaningful dimension, contributing to the broader conversation on safe and ethical AI. The findings open up new avenues for fine-tuned, targeted adjustments in diffusion models, making it easier for practitioners to achieve high-quality text rendering and control content generation more responsibly.

**Weaknesses:**

The method for identifying key cross and joint attention layers is central to the paper’s contribution, yet the description of this process could benefit from greater specificity. For instance, while the authors mention leveraging activation patching techniques, the exact steps involved in isolating specific layers and evaluating their effect on text generation are not fully detailed. Providing pseudocode or a workflow diagram would make the process clearer and more reproducible.
The approach is tested on models using CLIP and T5 encoders, yet diffusion models incorporating different or proprietary text encoders could present challenges to the proposed localization technique. Extending the evaluations to other popular encoders (or discussing limitations for certain encoder types) would improve the generalizability of the method.
While LoRA fine-tuning on localized layers is shown to enhance text generation quality, the paper does not deeply analyze potential trade-offs involved. Specifically, fine-tuning localized layers only may limit generalization to unseen prompts or introduce subtle biases in text rendering quality across different styles or languages. Including a comparative analysis of localized vs. whole-model fine-tuning would be helpful, especially with examples illustrating where one approach might be preferred over the other.
While the method is shown to be effective on certain models, scaling the approach for larger models or more computationally intensive tasks may present challenges. A more detailed discussion on these potential limitations would provide a balanced view and help guide future research in optimizing model efficiency.

**Questions:**

Could the authors provide more details on the method for identifying the critical cross and joint attention layers? Specifically, is the layer selection based solely on empirical observations, or are there theoretical underpinnings guiding this choice?

Have the authors tested whether fine-tuning only the localized layers with LoRA introduces any long-term risks, such as overfitting or loss of generalization? Are there any benchmarks or datasets where this approach might underperform relative to fine-tuning all cross-attention layers?

How does the method handle diffusion models that utilize text encoders other than CLIP and T5, or models with proprietary modifications? Does the localization process adapt effectively to these scenarios, or are there specific challenges?

Why were SimpleBench and CreativeBench chosen as the primary benchmarks? Would the method be equally effective on more complex, real-world benchmarks that include diverse text styles, such as cursive or stylized fonts?

Could the approach for localizing parameters influencing text generation extend to other applications? The paper would benefit from a discussion of possible applications beyond text generation and toxicity prevention, potentially sparking ideas for future work or new applications of the method.

---

> ### Author Response · Authors · 2024-11-22
> **Response to the Review**
>
> We thank to the reviewer for appreciating the innovation and novelty of our work and valuable feedback. Below we address the main concerns raised in the review:
>
> > The method [...] is central to the paper’s contribution, yet the description of this process could benefit from greater specificity. [...] the exact steps involved in isolating specific layers and evaluating their effect on text generation are not fully detailed. Providing pseudocode or a workflow diagram would make the process clearer and more reproducible.
>
> > Specifically, is the layer selection based solely on empirical observations, or are there theoretical underpinnings guiding this choice?
>
> We are grateful to the Reviewer for noticing that the process of finding layers in our work needs further clarification. As suggested in Appendix, Section A.9, we present in Algorithm 1 the pseudocode for finding a set of layers that we call $L_{ours}$ responsible for generating textual content. This procedure is applied in our work to all three analyzed diffusion models.
>
> Based on the provided pseudocode, we hope that the process is clear. However, we are more than happy to answer any further questions.
>
> > Diffusion models incorporating different or proprietary text encoders could present challenges to the proposed localization technique. Extending the evaluations to other popular encoders (or discussing limitations for certain encoder types) would improve the generalizability of the method. [...] How does the method handle diffusion models that utilize text encoders other than CLIP and T5, or models with proprietary modifications? Does the localization process adapt effectively to these scenarios, or are there specific challenges?
>
> We successfully perform our localization experiments on three large-scale diffusion models using different combinations of diverse text encoders. To be more precise, Stable Diffusion XL leverages two CLIP models (CLIP ViT-L & OpenCLIP ViT-bigG), DeepFloyd IF - one T5 model (T5-XXL) and Stable Diffusion 3 -- two CLIP models (CLIP ViT-L & OpenCLIP ViT-bigG) and T5 model (T5-XXL). Other large-scale diffusion models capable of high-quality text generation - Imagen (using T5-XXL) and FLUX (using CLIP L/14 and T5-XXL) -- also leverage combinations of those two encoders.
>
>
> It is worth noting that T5 and CLIP text encoders are very different architectures. While T5 leverages a bidirectional attention mechanism, CLIP encoders are based on causal attention (propagating tokens in only one direction). We therefore conclude that our experiments are, in fact, broad in scope, as we show that our approach spans across several conceptually different large-scale diffusion models.
>
> > While LoRA fine-tuning on localized layers is shown to enhance text generation quality, the paper does not deeply analyze potential trade-offs involved. Specifically, fine-tuning localized layers only may limit generalization to unseen prompts or introduce subtle biases in text rendering quality across different styles or languages. Including a comparative analysis of localized vs. whole-model fine-tuning would be helpful, especially with examples illustrating where one approach might be preferred over the other. While the method is shown to be effective on certain models, scaling the approach for larger models or more computationally intensive tasks may present challenges. A more detailed discussion on these potential limitations would provide a balanced view and help guide future research in optimizing model efficiency.
>
> We fine-tune localized layers on a subset of data from the MARIO-10M benchmark and evaluate the model on a distinct test set consisting of prompts from SimpleBench and CreativeBench, which were unseen during training. This evaluation assesses the model's generalization ability. Our experiments demonstrate that fine-tuning localized layers has minimal impact on the diversity and fidelity of the generations compared to the base model. We measure this by calculating precision and recall metrics throughout the fine-tuning process, as shown on the left side of Figure 5 in our submission.
>
> While it is true that we did not validate the model's quality across different styles, languages, or types of text, the MARIO-10M benchmark primarily comprises images from the LAION-400M [1] dataset and The Movie Database (TMDB). This makes it a highly diverse dataset containing texts in various styles. Therefore, fine-tuning localized layers does not significantly limit the base model's ability to generate diverse text.
>
> [1] Schuhmann, Christoph, et al. "Laion-400m: Open dataset of clip-filtered 400 million image-text pairs." arXiv preprint arXiv:2111.02114 (2021).

---

> > ### Author Response · Authors · 2024-11-22
> > **Response to the Review**
> >
> > > Have the authors tested whether fine-tuning only the localized layers with LoRA introduces any long-term risks, such as overfitting or loss of generalization? Are there any benchmarks or datasets where this approach might underperform relative to fine-tuning all cross-attention layers?
> >
> > As presented in Figure 5 of our submission, fine-tuning only the localized layers with LoRA introduces a very small decrease in the general performance of the model as measured by Precision and Recall. This is in contrary to the fine-tuning of all cross-attention layers, where we can observe higher overfitting. To answer this question further, we ran an additional experiment that we included in Appendix A8. There, we show that our approach with single-layer fine-tuning outperforms full fine-tuning in additional scenarios with various amounts of data samples.
> >
> > > Why were SimpleBench and CreativeBench chosen as the primary benchmarks?
> >
> > In our experiments, we follow benchmarks introduced with recent work by Yang et al. (NeurIPS 2023) [1] whose goal was to *increase text generation capabilities* of DMs. Thus, the benchmarks assess accurately the performance of text-to-image models in generating visually accurate text.
> >
> > > Would the method be equally effective on more complex, real-world benchmarks that include diverse text styles, such as cursive or stylized fonts?
> >
> > In our evaluation, we focus solely on the quality of the generated text and the background. The evaluation is performed entirely with the generations and does not require access to any real-world images. Therefore, it will not benefit from more complex benchmarks. Nevertheless, we are grateful for your suggestions to look into additional features of generated texts. We run additional experiments that evaluate how our approach can be extended to this setup. In Appendix A.9 of the updated submission, we include the study on the localization of parameters controlling the style of the generated text. Our experiments indicate that the layer we localize for textual content does not control the style of the generated text. Furthermore, we attempt to find the parameters responsible for this text property and show that the style is controlled by at least seven layers of cross-attention in Stable Diffusion 3. This indicates that not all visual aspects of an image can be controlled using a single layer, making our contribution more unique.
> >
> > > Could the approach for localizing parameters influencing text generation extend to other applications? The paper would benefit from a discussion of possible applications beyond text generation and toxicity prevention, potentially sparking ideas for future work or new applications of the method.
> >
> > Thank you for this valuable suggestion. Even though the main contribution of our work is the localization of a small part of the model responsible for the generation of textual content, we also introduced three applications where our observations can be used in practice. However, we agree that the list of potential benefits of our findings is not limited to the evaluated use cases. In particular, we think that we can easily adapt our method to the task of text removal or text addition, including an extension to other textual features such as styles or fonts for which we included initial experiments in the Appendix. As an interesting future work, we envision analyses in other domains, such as video generation, where layer specialization might be especially valuable considering the problem of consistency between video frames.
> >
> > **References:**
> >
> > [1] *"GlyphControl: Glyph Conditional Control for Visual Text Generation"* Yukang Yang, Dongnan Gui, Yuhui Yuan, Weicong Liang, Haisong Ding, Han Hu, Kai Chen. NeurIPS 2023.

---

> > > ### Author Response · Authors · 2024-12-02
> > > **Official Comment by Authors**
> > >
> > > Dear Reviewer mH7c, with the discussion period concluding on December 2nd AoE, we would appreciate you reviewing our responses and confirming whether they adequately address your questions and concerns about the submission.

---

### Official Review · Reviewer_9sEa · 2024-11-01

**Soundness:** 3
**Presentation:** 3
**Contribution:** 3
**Rating:** 6
**Confidence:** 4

**Summary:**

This paper explores to find a small subset of attention layers in DMs which determines text generated in images. It finds only a few layers related to this. Based on this observation, this paper further introduces a new fine-tuning strategy and a text edit technique. It can be used to prevent the generation of harmful or toxic text in images. Extensive experiments and analysis are conducted to demonstrate the observation and effectiveness of the proposed method.

**Strengths:**

The paper is well-structured, with a logical progression from the introduction to the conclusions. The writing is clear and concise, allowing complex exploration to be conveyed effectively. Key terms are introduced and defined appropriately, enhancing readability and ensuring that readers can follow the arguments easily. This observation is interesting and can effectively help us understand the inherent properties of the DMs in textual generation.

**Weaknesses:**

1、	The generalizability of the observations proposed in the paper needs further exploration. For instance, it remains to be determined whether this property holds consistently across texts of varying lengths and images with different aspect ratios.

2、	Different attention layers tend to focus on varying aspects, such as content or style. This has already been explored in previous works. This paper applies this feature to textual generation, yet more clarification is needed to highlight the novelty of this contribution.

3、	From some visualizations, it appears that the proposed approach not only alters the content of the text but also changes its style, such as the font. Are both style and content localized within the same layer?

**Questions:**

Hope the author can answer the questions in the weaknesses part.

---

> ### Author Response · Authors · 2024-11-22
> **Response to the Review**
>
> We thank the reviewer for finding our insights interesting and for the valuable feedback with concerns we wish to address.
>
> > The generalizability of the observations proposed in the paper needs further exploration. For instance, it remains to be determined whether this property holds consistently across texts of varying lengths and images with different aspect ratios.
>
> We appreciate the suggestion to test how our solution generalizes for longer-text generations and images with different aspect ratios. We show the results of those experiments below and include them in the appendix.
>
> First, using the Stable Diffusion 3 model, we recalculate results on both **Simple Bench** and **Creative Bench** using different aspect ratios. We show no significant differences in text edition performance between the set-up measured in our work (512x512) and other image resolutions (1024x1024, 1024x512, 512x1024). In particular, we find that, for images with aspect ratios of 1:2 and 2:1, our method tends to obtain better alignment (in OCR F1-Score) between the generated visual text and the one specified in the prompt. We can also observe improved background preservation capabilities for high-resolution images.
>
> **Simple Bench**
> | Setup    | MSE $\downarrow$ | SSIM $\uparrow$| PSNR  $\uparrow$| OCR F1  $\uparrow$| CLIP-T $\uparrow$ | LD $\downarrow$ |
> | -------- | ----- | ----- | ------ | -------- | -------- | ---- |
> | SD3 512x512 | 71.60 | 0.61 | 29.72 | 0.64 | 0.75 | 5.25 |
> | SD3 1024x1024 | 70.89 | 0.72 | 29.84 | 0.53 | 0.70 | 5.79
> | SD3 512x1024 | 66.22 | 0.67 | 30.12 | 0.68 | 0.73 | 4.24|
> | SD3 1024x512 | 70.68 | 0.65 | 29.85 | 0.56 | 0.71 | 6.33 |
>
> **Creative Bench**
> | Setup    | MSE $\downarrow$ | SSIM $\uparrow$| PSNR $\uparrow$ | OCR F1 $\uparrow$ | CLIP-T $\uparrow$ | LD $\downarrow$ |
> | -------- | -------- | -------- | -------- | -------- | -------- | -------- |
> | SD3 512x512 | 69.02 | 0.63 | 30.06 | 0.45 | 0.80 | 32.77 |
> | SD3 1024x1024 | 63.13 | 0.73 | 30.61 | 0.41 | 0.75 | 42.52 |
> | SD3 512x1024 | 67.85 | 0.69 | 30.05 | 0.47 | 0.79 | 29.76 |
> | SD3 1024x512 | 67.15 | 0.67 | 30.08 | 0.46 | 0.79 | 39.78 |
>
> We conducted an additional study with the SD3 model, measuring how our localization-based text edition performs across texts with multiple words. We present the results of this comparison in the table below. We note that editing longer texts results in slightly higher values of the background preservation loss, which is understandable given that proportionally longer texts usually occupy more space in the image. When it comes to the alignment between the generated text and the target prompt, we notice slightly worse performance for a larger number of words. Although the OCR F1-Score increases with the number of words, this metric is not comparable for texts of different lengths, and we find CLIP-T to be more reliable in this case. In Appendix A.11, we present example results of editing text for images with varying text lengths. The generations indicate that our method is also applicable in such a scenario.
>
> | Number of words | MSE  $\downarrow$ | SSIM $\uparrow$ | PSNR $\uparrow$ |  OCR F1 $\uparrow$  | CLIP-T  $\uparrow$ |
> | -- | ----- | ----- | ----- | -----  | ----- |
> | 1  | 0.677 | 0.695 | 0.302 | 0.377 | 0.746 |
> | 2  | 0.706 | 0.675 | 0.300 | 0.403 | 0.717 |
> | 3  | 0.703 | 0.676 | 0.300 | 0.442 | 0.721 |
> | 4  | 0.725 | 0.668 | 0.298 | 0.457 | 0.714 |
> | 5  | 0.726 | 0.664 | 0.298 | 0.474 | 0.698 |
> | 6  | 0.718 | 0.663 | 0.299 | 0.487 | 0.701 |
> | 7  | 0.724 | 0.654 | 0.298 | 0.489 | 0.704 |
> | 8  | 0.735 | 0.653 | 0.297 | 0.494 | 0.695 |

---

> > ### Author Response · Authors · 2024-11-22
> > **Response to the Review**
> >
> > > Different attention layers tend to focus on varying aspects, such as content or style. This has already been explored in previous works. This paper applies this feature to textual generation, yet more clarification is needed to highlight the novelty of this contribution.
> >
> > Our work is based on previous studies that tackled the problem of layer specialization. We overview such publications in the section "Interpretability of diffusion models" of the related work. Comparing to those approaches, we provide several new insights that were not studied before:
> >
> > - To the best of our knowledge, this is the first work focusing on the localization of layers responsible for textual content generation.
> > - Contrary to recent studies, we extend our analyses over several large-scale models with various architectures (U-Net, Diffusion Transformers), training paradigms (diffusion, flow matching), and different text encoders (CLIP, T5). We show that our hypothesis holds independently of those design choices.
> > -  On top of our main observation, we introduce several applications that directly benefit from the localization of text-specific layers. This includes parameter-efficient fine-tuning and safeguarding text-to-image models, which, to the best of our knowledge, were not associated with layers specialization before.
> >
> > Even though this submission does not include a new algorithm, we still believe we have brought insights valuable to the ML community.
> >
> > >From some visualizations, it appears that the proposed approach not only alters the content of the text but also changes its style, such as the font. Are both style and content localized within the same layer?
> >
> > We thank the reviewer for this suggestion, which poses an interesting research question. For all of our experiments in the original submission, we did not specify the style of the text in the source or the target prompt. Hence, we did not expect the textual style to be preserved. However, in Appendix A.9 of the updated submission, we conduct an additional experiment showing that the generated text style is not localized together with its content within the same layer. In the case of Stable Diffusion 3, textual style is, in fact, controlled by at least seven layers of cross-attention, which indicates that not all the visual aspects of an image can be controlled using a single layer.

---

> > > ### Comment · Reviewer_9sEa · 2024-11-30
> > >
> > > Thanks for the detailed response and additional experiments. I believe that the exploration of this article can provide more insights for the community, and I still believe that more exploration is needed for the control of style and content .To this end, I'd like to keep my rating.

---

### Official Review · Reviewer_aXtp · 2024-11-03

**Soundness:** 3
**Presentation:** 4
**Contribution:** 3
**Rating:** 8
**Confidence:** 4

**Summary:**

This paper made an interesting observation on how visual text specified for text-to-image generation is realized through the cross-attention layers - there are only a few cross-attention layers responsible for processing the information about the text information. This is identified by swapping the key and values to be generated across every layer. With such observation, the authors perform several experiments showing the effectiveness of using these layers for different visual text applications.

**Strengths:**

[Originality]

The finding about the function of cross-attention layers in generating visual text is quite novel and interesting.

[Quality]

The authors did rigorous experiments to show that only a few layers are responsible for generating visual texts with multiple experiment setups. Additionally, the authors show that these layers only focus on these visual texts instead of generating other content described by the text prompts. The applications of improving text generation, editing, and preventing the generation of toxic text also demonstrate the usefulness of the findings.


[Clarity]

The paper did a very good job of presenting the analysis and applications.

[Significance]

I personally find that this is an intriguing finding about the interpretation of the text-to-image diffusion model. The paper paves a way towards understanding how text embeddings are used through attention layers to control the text-to-image model. The finding also shows promise for several downstream tasks.

**Weaknesses:**

I did not find significant weaknesses in the paper. However, the authors could consider the following feedback to improve the readiness and clarity of the paper:

1. Although I'm fairly familiar with the architectures of SD3 and SDXL, I still need to guess how localizing by patching is different from injection. So, for SD3, are only the keys and values corresponding to text embeddings swapped by the target prompt caching?

2. There are many papers in text-to-image generation/editing working on cross-attention layers and manipulating the denoising steps through keys and values. The authors should consider doing more comprehensive survey and here are some of them:

[1] Kumari, Nupur, et al. "Multi-concept customization of text-to-image diffusion." Proceedings of the IEEE/CVF Conference on Computer Vision and Pattern Recognition. 2023.

[2] Geyer, Michal, et al. "TokenFlow: Consistent Diffusion Features for Consistent Video Editing." The Twelfth International Conference on Learning Representations.

[3] Patashnik, Or, et al. "Localizing object-level shape variations with text-to-image diffusion models." Proceedings of the IEEE/CVF International Conference on Computer Vision. 2023.

[4] Phung, Quynr, et al. "Grounded text-to-image synthesis with attention refocusing." Proceedings of the IEEE/CVF Conference on Computer Vision and Pattern Recognition. 2024.

[5] Tumanyan, Narek, et al. "Plug-and-play diffusion features for text-driven image-to-image translation." Proceedings of the IEEE/CVF Conference on Computer Vision and Pattern Recognition. 2023.

**Questions:**

Minor questions and suggestions:

1. Is there any specific reason to classify the target text into different buckets and use the most frequent texts as the validation set? It seems other buckets are not used.

2. Lines 144 and 145 look like new paragraph names.

---

> ### Author Response · Authors · 2024-11-22
> **Response to the Review**
>
> We thank the Reviewer for recognizing the quality and significance of our work.
>
> > Although I'm fairly familiar with the architectures of SD3 and SDXL, I still need to guess how localizing by patching is different from injection. So, for SD3, are only the keys and values corresponding to text embeddings swapped by the target prompt caching?
>
> We are grateful to the reviewer for pointing out a hard-to-understand part of our submission. We have described the differences between injection and patching in lines 213-235 of Section 4.1 and in Figure 1 of the original submission, but following the reviewer's suggestions, we improved it in the updated version of the paper. Below, we additionally explain the main differences in more detail.
>
> In architectures based on UNet (DeepFloyd IF and SDXL), text embeddings produced by the text encoder are directly multiplied by key and value matrices in each cross-attention layer. Thus, as each cross-attention layer receives exactly the same text embedding, in the *injection* approach, we can simply change the input text embedding for each of the analyzed cross-attention layers.
> However, in architectures that are Multimodal Diffusion Transformers (SD3) leveraging the Joint Attention mechanism, only the very first layer of the diffusion model receives the direct text encoder's embedding. Every next layer receives embedding **processed by preceding layers** of a transformer. Hence, it is necessary to swap text keys and values corresponding to the source prompt with previously cached keys and values calculated for the target one. Importantly, as correctly noticed by the reviewer, we cache only keys and values corresponding to the processing of *text* embeddings (in each cross-attention layer in SD3, there is another set of keys and values that process image embeddings).

---

> > ### Author Response · Authors · 2024-11-22
> > **Response to the Review**
> >
> > > There are many papers in text-to-image generation/editing working on cross-attention layers and manipulating the denoising steps through keys and values. The authors should consider doing more comprehensive survey and here are some of them:
> >
> > We appreciate the Reviewer for highlighting these important works. We already included them in the updated version of our submission. Additionally, we provide a brief review below to illustrate their connection to our submission. While highly relevant, we note that these works are not directly comparable to our approach.
> >
> > _**Multi-Concept Customization of Text-to-Image Diffusion**_
> > In this work, the authors present an efficient way of customization of text-to-image diffusion models by fine-tuning a subset of cross-attention layer parameters. While their approach demonstrates that targeting the key and value matrices in all the cross-attention layers is sufficient to introduce new concepts, we reveal that fine-tuning those matrices in fewer than 5% of cross-attention layers (see Table 1 of submission) enables better quality of the generated text.
> >
> > _**TokenFlow: Consistent Diffusion Features for Consistent Video Editing**_
> > TokenFlow presents a framework that enables video editing using text-to-image diffusion models. Specifically, the authors introduce a method of editing the keyframes by extension of self-attention layers in which the keys from all timeframes are concatenated in order to encourage the frames to share a global appearance. The presented solution offers an effective approach to the semantic video edition, that's why it significantly differs from our work which targets image editing.
> >
> > _**Localizing object-level shape variations with text-to-image diffusion models**_
> > Prompt-Mixing enables users to explore different shapes of objects in an image. In order for objects to stay in the same positions but change their appearance, the method operates in the inference time and, in different denoising timestep intervals, injects different prompts into the cross-attention layers. In our work, we use a similar injection mechanism that we apply only to the selected text-controlling layers. We evaluate the effect of injection at different denoising steps in section A.2 in the Appendix, where we show that starting the patching from later timestep can positively affect both the quality of generated text and better preserve other visual attributes.
> >
> > _**Grounded text-to-image synthesis with attention refocusing**_
> > Cross-Attention Refocusing (CAR) is a calibration technique enabling better attending of tokens representing objects to image regions. By performing multiple intermediate latent optimizations by using CAR loss and Self-Attention Refocusing loss, authors achieve better controllability of the layout of generated objects. Similar to our work, CAR focuses on cross-attention maps but aims to strengthen attention to the correct token while reducing it elsewhere.
> >
> > _**Plug-and-play diffusion features for text-driven image-to-image translation**_
> > Plug-and-Play is an effective image-to-image translation method. In this work, authors show that in the denoising procedure, one can extract spatial features from the U-Net decoder's Residual Blocks and their following self-attention layers, obtaining encodings of the composition of the image. Next, by passing a different prompt during the denoising procedure for the same initial Gaussian noise, one can inject previously extracted features and obtain generations, differing in image attributes specified in the condition. In this work, we show that by focusing on text-related features, we can perform a precise edition by targeting a single attention layer.
> >
> > > Is there any specific reason to classify the target text into different buckets and use the most frequent texts as the validation set? It seems other buckets are not used.
> >
> > In our experiments, we use all four out of four buckets from both benchmarks (*SimpleBench* and *CreativeBench*). For fairness of results, we use a bucket of the most frequent words as a validation set: on this set, we localize attention layers responsible for the content of the generated text and find the starting timesteps $t_s$ for patching. Then, we evaluate the performance of our method on the remaining three buckets, which constitute the test set. We described the details of these splits in lines 146-147 of our submission. We also note that we evaluate our fine-tuning and editing performance on *test split* in lines 349 and 408, respectively.
> >
> > > Lines 144 and 145 look like new paragraph names.
> >
> > We thank the reviewer for pointing it out. We updated the text in this paragraph to eliminate confusion.
> >
> > We are delighted that you found our submission worthy of acceptance. We would be happy to address any further questions and hope for your continued support during the discussion period.

---

> > > ### Comment · Reviewer_aXtp · 2024-11-24
> > >
> > > Thanks for the response! I carefully read the other reviewers' concerns about the paper and connected with their perspectives on the fine-tuning experiments and applications. However, I still think the findings of this paper to be interesting and useful and could motivate more analysis works in this thread. To this end, I'd like to keep my rating.

---

> > > > ### Author Response · Authors · 2024-12-02
> > > > **Thank you!**
> > > >
> > > > Thank you for taking the time to thoroughly read our paper and carefully consider the other reviews, we appreciate it. Your high score and support for our work are incredibly motivating and inspire us to continue advancing in this area.

---

### Official Review · Reviewer_TSpC · 2024-11-03

**Soundness:** 3
**Presentation:** 3
**Contribution:** 2
**Rating:** 6
**Confidence:** 4

**Summary:**

This paper proposes to localize the attention layers which influence the text-rendering ability of tex-to-image generation models. With the localized layers, the authors propose to utilize them in many tasks including generation and editing.

**Strengths:**

Distribution of image samples with text may be quite different from image samples without text. As a result, we may face a trade-off between image quality and text-rendering ability in practice when training text-to-image generation model.
Thus the idea of localizing corresponding layers for text rendering ability in text-to-image generation models is interesting.
If we can localize such layers, then we can use carefully designed fine-tuning so that the resulting model performs well in both image quality and text accuracy.



From the experiments, we can see that the localized parameters indeed influence the text-rendering ability.

**Weaknesses:**

The fine-tuning experiment is conducted on a small subset of MARIO-10M dataset. So it is expected that fine-tuning the whole model may lead to overfitting. The experiment results can show fine-tuning localized layers indeed works, but it can not show that fine-tuning only localized layers is better than fine-tuning the whole model.
To illustrate the effects of the localized layers, it is suggested to conduct the experiment on large-scale dataset, i.e. fine-tuning the localized layers or whole model on all samples from MARIO-10M dataset.

Another case which is more related to the fine-tuning experiment with smaller dataset is fine-tuning the model on complicated samples of high-quality. Different from MARIO-10M which may contain some simple and low-quality samples, high-quality samples with long sentences rendered might be difficult to collect. It can strength the contribution of the proposed method if it can improve model performance under this setting.


The author mentioned "We focus on LoRA for SDXL since this model has a significantly lower text generation
quality compared to other studied DMs". Can fine-tuned SDXL outperform SOTA methods on generation task with commonly used benchmark? If it can not, why not fine-tuning one of the SOTA models to show that the proposed method also works and can be used to improve not only SDXL? Only fine-tuning a model with lower performance is not convincing.

More qualitative results (especially with more words to be generated) and human evaluation are suggested.

**Questions:**

Since MARIO-10M dataset is utilized in training, why is MARIO-Eval Benchmark not directly used to evaluate the generation performance in the paper?

What is the reason that the authors only use a small subset of Mario-10M dataset?

How is the editing performance compared to other methods like textDiffuser?

---

> ### Author Response · Authors · 2024-11-22
> **Response to the Review**
>
> > The fine-tuning experiment is conducted on a small subset of MARIO-10M dataset. So it is expected that fine-tuning the whole model may lead to overfitting. The experiment results can show fine-tuning localized layers indeed works, but it can not show that fine-tuning only localized layers is better than fine-tuning the whole model. To illustrate the effects of the localized layers, it is suggested to conduct the experiment on large-scale dataset, i.e. fine-tuning the localized layers or whole model on all samples from MARIO-10M dataset.
>
> > What is the reason that the authors only use a small subset of Mario-10M dataset?
>
> We thank the Reviewer for suggesting the fine-tuning experiment evaluation on a larger-scale dataset. We consider it particularly interesting for the setup where all cross-attention layers are fine-tuned.
>
> For additional evaluation, we extended the training set from 78k to 200k samples drawn from the MARIO-10M dataset. While fine-tuning all cross-attention layers on the entire 10M samples in the MARIO benchmark is infeasible within the rebuttal period, this more than twofold increase in training data allows us to approximate the impact of scaling up the dataset. Additionally, to assess the effect of training set size on the fine-tuning of only localized layers, we conducted experiments using training sets ranging from 20k to 200k samples for this setup. Each configuration was trained for 12k steps with a batch size of 512.
>
> All hyperparameters were kept consistent across setups, with the only variable being the number of unique training samples. The table below presents the final metrics for each model. Further experimental details and metrics recorded during training are included in Appendix A.8 of our updated submission.
>
>
> | Setup | # of train samples | Recall $\uparrow$ | Precision $\uparrow$ | OCR F1 $\uparrow$ | CLIP-T $\uparrow$ | Training time [h] |
> | -------- | -------- | -------- | -------- | -------- | -------- | -------- |
> | Base model (before fine-tuning) | -     |  -    | - | 0.433 | 0.826 | - |
> | Full model | 78k                 |  0.728    | 0.785 | 0.497 | 0.833 | 25.5h |
> | Full model | 200k                |  0.779   | 0.777 | 0.506 | 0.831 | 25.5h |
> | Ours localized layers | 20k     |   **0.997**  | **0.995** | **0.561** | **0.854** | 18.5h |
> | Ours localized layers | 50k     |   0.899  | 0.967 | 0.556 | 0.853 | 18.5h |
> | Ours localized layers | 78k     |   0.946  | 0.986 | 0.554 | **0.854** | 18.5h |
> | Ours localized layers | 150k     |   0.894  | 0.981 | 0.549 | 0.852 | 18.5h |
> | Ours localized layers | 200k     |   0.902  | 0.971 | 0.555 |0.852  | 18.5h |
>
> Increasing the training set size improves the performance when all cross-attention layers are fine-tuned. However, this approach still notably underperforms compared to fine-tuning only localized cross-attention layers. Interestingly, all setups involving localized fine-tuning achieve comparable performance, regardless of the variation in training set size. This underscores that targeting layers specifically responsible for textual generation enables effective fine-tuning to enhance text generation quality, even with a very limited number of training samples.
>
>
>
> > Another case which is more related to the fine-tuning experiment with smaller dataset is fine-tuning the model on complicated samples of high-quality. Different from MARIO-10M which may contain some simple and low-quality samples, high-quality samples with long sentences rendered might be difficult to collect. It can strength the contribution of the proposed method if it can improve model performance under this setting.
>
>
> We evaluated how the characteristics of samples, particularly the length and quality of text, impact fine-tuning performance. To that end, we sorted all samples in the MARIO-10M dataset based on text length and OCR F1 scores, as measured by our OCR model. From this sorted dataset, we selected the top 1k samples, which represent the most complex and potentially most useful cases, that we used for fine-tuning. Below, we compare the results of this experiment with the fine-tuning on 1k of randomly selected samples.
>
> | Setup | OCR F1 | CLIP-T |
> | -------- | -------- | -------- |
> | Base model (before fine-tuning) | 0.433 | 0.826 |
> | Random 1k samples     | **0.566**     | **0.855**  |
> | Top 1k sorted samples     | 0.556     | 0.854     |
>
> Our results show no significant difference in performance between the two setups. This suggests that, even with a very small dataset, the complexity or quality of the samples used for fine-tuning does not have a notable impact on performance.

---

> > ### Author Response · Authors · 2024-11-22
> > **Response to the Review**
> >
> > > The author mentioned "We focus on LoRA for SDXL since this model has a significantly lower text generation quality compared to other studied DMs". Can fine-tuned SDXL outperform SOTA methods on generation task with commonly used benchmark? If it can not, why not fine-tuning one of the SOTA models to show that the proposed method also works and can be used to improve not only SDXL? Only fine-tuning a model with lower performance is not convincing.
> >
> > We evaluate both the base SDXL model and the SDXL model fine-tuned with LoRA on a benchmark for text editing, as shown in Table 3 of our submission. The improvement in the OCR F1 metric for the LoRA fine-tuned model over the base model (0.43 vs. 0.34) is solely attributable to the fine-tuning of localized cross-attention layers, demonstrating its effectiveness.
> >
> > However, even the fine-tuned SDXL model achieves significantly lower results on the OCR F1 metric compared to the DeepFloyd IF (0.70) and SD3 (0.68) models. Following the reviewer's suggestion, we also attempted fine-tuning localized layers for these models, but we were not able to improve their performance. We hypothesize that this is due to the use of the T5 text encoder in both DeepFloyd IF and SD3, which proved to significantly enhance models' ability to generate high-quality text in images, as studied in [1].
> >
> >
> > [1] Esser, Patrick, et al. "Scaling rectified flow transformers for high-resolution image synthesis." Forty-first International Conference on Machine Learning. 2024.
> >
> > > More qualitative results (especially with more words to be generated) and human evaluation are suggested.
> >
> > To answer this question, we conducted an additional study with the SD3 model, measuring how our localization-based text edition performs across texts with multiple words. We present the results of this comparison in the table below. We note that editing longer texts results in slightly higher values of the background preservation loss, which is understandable given that proportionally longer texts usually occupy more space in the image. Regarding the alignment between the generated text and the target prompt, we notice slightly worse performance for a larger number of words. Although the OCR F1-Score increases with the number of words, this metric is not comparable for texts of different lengths, and we find CLIP-T to be more reliable in this case. In Appendix A.11, we present example results of editing text for images with varying text lengths. The generations indicate that our method is also applicable in such a scenario.
> >
> > | Number of words | MSE  $\downarrow$ | SSIM $\uparrow$ | PSNR $\uparrow$ |  OCR F1 $\uparrow$  | CLIP-T  $\uparrow$ |
> > | -- | ----- | ----- | ----- | -----  | ----- |
> > | 1  | 0.677 | 0.695 | 0.302 | 0.377 | 0.746 |
> > | 2  | 0.706 | 0.675 | 0.300 | 0.403 | 0.717 |
> > | 3  | 0.703 | 0.676 | 0.300 | 0.442 | 0.721 |
> > | 4  | 0.725 | 0.668 | 0.298 | 0.457 | 0.714 |
> > | 5  | 0.726 | 0.664 | 0.298 | 0.474 | 0.698 |
> > | 6  | 0.718 | 0.663 | 0.299 | 0.487 | 0.701 |
> > | 7  | 0.724 | 0.654 | 0.298 | 0.489 | 0.704 |
> > | 8  | 0.735 | 0.653 | 0.297 | 0.494 | 0.695 |
> >
> > > Since MARIO-10M dataset is utilized in training, why is MARIO-Eval Benchmark not directly used to evaluate the generation performance in the paper?
> >
> > Our evaluation of the model's fine-tuning does not require access to any real images, and therefore, we do not use images provided in the MARIO Eval benchmark. Instead, we rely solely on prompts, ensuring they are distinct from those used during fine-tuning. To measure the quality of the generated text, we utilize an external OCR to extract text from the generated images and calculate how closely it matches the text specified in the prompts.
> >
> > For this evaluation, we use a test set of 300 prompts sourced from both *SimpleBench* and *CreativeBench*. We selected these benchmarks to align our assessment with prior works aiming to improve text generation quality in text-to-image diffusion[1].
> >
> > [1] *"GlyphControl: Glyph Conditional Control for Visual Text Generation"* Yukang Yang, Dongnan Gui, Yuhui Yuan, Weicong Liang, Haisong Ding, Han Hu, Kai Chen. NeurIPS 2023.
> >
> > > How is the editing performance compared to other methods like textDiffuser?
> >
> > In the text edition experiment, we compare our method's performance against Prompt-to-Prompt, which, similarly to us, uses only source and target prompts to modify an image generated by the diffusion model. The TextDiffuser method strongly relies on the Stable Diffusion 1.5 backbone with an additional layout prediction model. At the same time, our work utilizes newer backbone models but does not require any additionally trained models that improve the quality of generated text. Therefore, we believe that there is no possibility of a fair comparison between those two techniques.
> >
> >
> > ------
> >
> > If there are any further questions we are more than happy to answer them. If there are not, in light of those clarifications, we ask the Reviewer to consider raising the score.

---

> > > ### Comment · Reviewer_TSpC · 2024-11-26
> > >
> > > The authors provided some analysis, which addressed my concerns especially on long text case. I would like to increase the score.

---

> > > > ### Author Response · Authors · 2024-12-02
> > > > **Thank you!**
> > > >
> > > > We appreciate the Reviewer's thoughtful feedback and are glad that the additional analysis on editing longer text addressed the concerns. Thank you for increasing your score and supporting our efforts on this topic.

---

### Official Review · Reviewer_RLCH · 2024-11-03

**Soundness:** 3
**Presentation:** 3
**Contribution:** 2
**Rating:** 5
**Confidence:** 3

**Summary:**

This paper discovered that only a small amount of parameters within modern diffusion models are in charge of textual content generation. This phenomenon is widely observed among U-Net-based diffusion models like SDXL and transformer-based ones like SD3. Based on this observation, this paper developed three possible application scenarios, including LoRA-based fine-tuning for enhanced text generation, textual content editing for images generated by diffusion models, and the prevention of toxic text generation.

**Strengths:**

- By careful experimental analysis, this paper localized a small subset of cross and joint attention layers in diffusion models that are responsible for textual content generation.
- Based on this observation, this paper developed an effective fine-tuning strategy for enhanced text generation while maintaining the model’s overall generation performance.

**Weaknesses:**

For the application of preventing the generation of toxic text within images, this paper claimed that their approach is able to remove the toxic text in the text prompt from the generated images. They achieved this target by first detecting the toxic text in the text prompt using state-of-the-art large language models, and then replacing the toxic text with non-harmful text for image generation using their approach. However, it seems that directly replacing the toxic text with non-harmful text and feeding the modified text prompt to the diffusion models can achieve the same target. Since the image generated by the source prompt won’t be provided to the users, there seems no need to prevent other visual content from being influenced when changing the source prompt (with toxic text) to the target prompt (without toxic text).

**Questions:**

Request the authors clarify the necessity of using their approach to prevent toxic textual content generation.

---

> ### Author Response · Authors · 2024-11-22
> **Response to the Review**
>
> We greatly appreciate the Reviewer's recognition of our analysis and valuable feedback regarding the preservation of toxic text generation.
>
> > it seems that directly replacing the toxic text with non-harmful text and feeding the modified text prompt to the diffusion models can achieve the same target. Since the image generated by the source prompt won’t be provided to the users, there seems no need to prevent other visual content from being influenced when changing the source prompt (with toxic text) to the target prompt (without toxic text).
>
> We are thankful for pointing out this simple baseline we omitted in our analysis. While we agree that, with safety mechanisms in place, the user will not be presented with the desired generation, we argue that at the same time, the generated image should align with the user's intention as closely as possible. Therefore, toxic words such as swearing, usually associated with intense emotions, should be reflected in the whole generation, even though we do not want to show them explicitly. To validate the point, we ran an additional experiment to measure a person's facial expression change when generated with a sign containing either toxic or benign text. We observed that thanks to the cross-attention mechanism, persons generated with toxic prompts appeared 10% angrier than those generated with the same prompt and from the same random seed but with benign texts on the sign, suggested as a replacement by the LLM. We did not observe such issues with our method when replacement texts were applied only to the selected layers. Please refer to Sec. A.6. of the appendix for detailed experiment results, including some generated examples.
>
>
> For completeness, we added this idea that we dubbed **Prompt Swap** to the general evaluation in the updated version of our submission. For convenience, we also show the updated results in this comment. We can observe that **Prompt Swap** is on par with our method when considering metrics related to the toxicity of the generated text, but it strongly affects other visual aspects of the generated image.
>
> We hope that our explanations and additional experiments are convincing and explain the potential of the last of the proposed applications. If you have any further questions, we are open to discussion. If not, we kindly ask the reviewer to consider raising the score.
>
>
> ### SDXL
> | Method              | MSE  $\downarrow$   | SSIM  $\uparrow$   | PSNR  $\uparrow$   | OCR F1 (Harmful) $\downarrow$ | Toxicity Score  $\downarrow$ |
> |---------------------|---------|----------|----------|----------|------------------|
> | Ours                | 48.20   | 0.79     | 31.68    | 0.20     | 0.003            |
> | Negative Prompt     | 77.95   | 0.71     | 31.76    | 0.23     | 0.052            |
> | Safe Diffusion      | 49.46   | 0.81     | 31.33    | 0.34     | 0.222            |
> | Safe Diffusion*     | 49.41   | 0.81     | 31.33    | 0.33     | 0.209            |
> | Prompt Swap         | 79.41   | 0.66     | 31.65    | 0.19     | 0.000            |
>
> ### DeepFloyd IF
> | Method              | MSE  $\downarrow$  | SSIM $\uparrow$  | PSNR  $\uparrow$  | OCR F1 (Harmful) $\downarrow$ | Toxicity Score $\downarrow$ |
> |---------------------|---------|----------|----------|----------|------------------|
> | Ours                | 74.96   | 0.61     | 29.60    | 0.32     | 0.018            |
> | Negative Prompt     | 100.50  | 0.37     | 28.12    | 0.59     | 0.250            |
> | Safe Diffusion      | 64.30   | 0.73     | 30.19    | 0.79     | 0.555            |
> | Safe Diffusion*     | 63.65   | 0.74     | 30.25    | 0.79     | 0.540            |
> | Prompt Swap         | 100.99  | 0.35     | 28.10    | 0.30     | 0.015            |
>
> ### SD3
> | Method              | MSE  $\downarrow$  | SSIM  $\uparrow$  | PSNR   $\uparrow$ | OCR F1 (Harmful) $\downarrow$ | Toxicity Score $\downarrow$ |
> |---------------------|---------|----------|----------|----------|------------------|
> | Ours                | 72.61   | 0.70     | 29.72    | 0.32     | 0.018            |
> | Negative Prompt     | 101.63  | 0.53     | 28.08    | 0.77     | 0.407            |
> | Safe Diffusion      | 34.99   | 0.86     | 34.25    | 0.73     | 0.571            |
> | Safe Diffusion*     | 33.67   | 0.87     | 34.56    | 0.73     | 0.568            |
> | Prompt Swap         | 98.58   | 0.51     | 28.22    | 0.30     | 0.015            |

---

> > ### Author Response · Authors · 2024-12-02
> > **Official Comment by Authors**
> >
> > Dear Reviewer RLCH, with the discussion period concluding on December 2nd AoE, we would appreciate you reviewing our responses and confirming whether they adequately address your questions and concerns about the submission.

---

> > > ### Comment · Reviewer_RLCH · 2024-12-03
> > >
> > > The additional experiments and detailed response are appreciated. While this paper advances our understanding of diffusion models, the necessity of their proposed approach for preventing toxic content generation remains questionable. The "Prompt Swap" method achieves comparable or superior results in terms of "OCR F1 (Harmful)" and "Toxicity Score" metrics. Furthermore, Figure 8 shows that the difference in "angry" scores between "Ours" and "Non-toxic" conditions is not substantial when compared to "Toxic" conditions. Moreover, users can still influence the emotional tone of generated images by adding terms like "angry" to prompts outside the "text" portion. Given these considerations, I would like to maintain my original rating.

---

### Author Response · Authors · 2024-11-27
**General response by Authors**

We would like to express our sincere gratitude to all the reviewers for their valuable feedback and insightful suggestions, which have enabled us to improve our work further. We are pleased to see that most reviewers found the findings in our submission novel and interesting and are inclined to accept it.

Reviewers found our *"idea of localizing corresponding layers for text rendering ability in text-to-image generation models"* interesting (Reviewers TSpC and 9sEa), innovative (Reviewer mH7c), novel, and intriguing (Reviewer aXtp). This unique work *"paves a way towards understanding how text embeddings are used through attention layers to control the text-to-image model and shows promise for several downstream tasks*" (Reviewer aXtp). *"The method is broadly applicable across various architectures, making it an impactful step forward in the interpretability and control of DMs"* (Reviewer mH7c). The experiments were carried out carefully (Reviewer RLCH), rigorously (Reviewer aXtp), and extensively (Reviewer 9sEa) to show that only a few layers are responsible for generating the visual text. The paper was clearly written and well-structured (Reviewer 9sEa), and *"did a very good job of presenting the analysis and applications"* (Reviewer aXtp).


Based on the reviewers' comments, we conducted the following experiments to improve our submission:

- **Toxic text removal.** We conducted an experiment demonstrating that our text-editing method effectively removes toxic content from images while preserving user-specified emotions in the prompt. This makes our approach more practical than simple baselines, such as replacing toxic words in the prompt and passing them to all the layers. The results are detailed in **Sec. A.6** of the Appendix.

- **Fine-tuning experiments.** We extended our fine-tuning experiments to varying training set sizes. We show that fine-tuning all cross-attention layers underperforms compared to fine-tuning only localized layers, even with a significant increase in training set size. Furthermore, fine-tuning localized layers achieves consistent performance across different training set sizes, demonstrating robustness. These results are presented in **Sec. A.8** of the Appendix.

- **Multi-word text editing:** We evaluated our method on images with texts containing multiple words. Although text alignment slightly decreases as the number of words increases, our method successfully edits text while preserving visual attributes. Results and examples are provided in **Sec. A.11** of the Appendix.

- **Localization of text style.** We conducted a study showing that control of text style, contrary to its content, is distributed across multiple cross-attention layers in diffusion models. The results of this study are presented in **Sec. A.9** of the Appendix.

On top of that, we improved the clarity of the paper, addressing some reviewers' concerns. In Section A.9 of the Appendix, we included **pseudocode for localizing the subset of cross-attention layers that control visual text**. In Section A.1  of the Appendix, we reviewed **related work** on manipulating diffusion models with cross-attention layers, indicating how those works differ from our approach. Additionally, we improved Section 4.1 to clarify details of the **patching technique** and how it differs from the injection.

We strongly believe that the revised version of our work presents a compelling contribution, offering valuable insights for the community. It advances the understanding of large-scale neural architectures and supports the development of novel methods for fine-tuning, image editing, and mitigating toxic content in generative models.

---

### Meta-Review · Area_Chair_GFLQ · 2024-12-18

**Metareview:**

This paper identifies a small subset of attention layers in diffusion models that drives textual generation. Based on this finding, the authors explore efficient fine-tuning to enhance text generation, text editing, and toxic content prevention. Experiments demonstrate improved text alignment and visual consistency.

After the discussion between the authors and reviewers, 4 out of 5 reviewers leaned toward acceptance. One reviewer expressed concerns about the performance of this work on a downstream task, specifically preventing toxic content generation. However, the strengths of this submission outweigh its limitations, making it worthy of publication at ICLR. The authors are encouraged to provide additional experimental results on preventing toxic content generation.

**Additional Comments On Reviewer Discussion:**

Four out of five reviewers participated in the authors-reviewers discussion, and one reviewer increased their rating.

---

### Decision · Program_Chairs · 2025-01-22

Accept (Poster)